# Finite Amplitude Stability of Internal Steady Flows of the Giesekus Viscoelastic Rate-Type Fluid

**DOI:** 10.3390/e21121219

**Published:** 2019-12-13

**Authors:** Mark Dostalík, Vít Průša, Karel Tůma

**Affiliations:** Faculty of Mathematics and Physics, Charles University, Sokolovská 83, Praha 8 – Karlín, 18675 Prague, Czech Republic; dostalik@karlin.mff.cuni.cz (M.D.); ktuma@karlin.mff.cuni.cz (K.T.)

**Keywords:** viscoelastic rate-type fluids, nonlinear stability, Lyapunov functional, thermodynamics, thermodynamically open systems, Bures distance, Wasserstein metric, 35Q79, 37L15, 37B25

## Abstract

Using a Lyapunov type functional constructed on the basis of thermodynamical arguments, we investigate the finite amplitude stability of internal steady flows of viscoelastic fluids described by the Giesekus model. Using the functional, we derive bounds on the Reynolds and the Weissenberg number that guarantee the unconditional asymptotic stability of the corresponding steady internal flow, wherein the distance between the steady flow field and the perturbed flow field is measured with the help of the Bures–Wasserstein distance between positive definite matrices. The application of the theoretical results is documented in the finite amplitude stability analysis of Taylor–Couette flow.

## 1. Introduction

Flows of viscoelastic fluids exhibit the phenomenon dubbed “elastic turbulence” or “inertia-less turbulence”. The flows of viscoelastic fluids can become—unlike the flows of the standard viscous fluids—unstable or “turbulent” at very low values of the Reynolds number. This behaviour indicates that the instability or transition to “turbulence” is driven by a nonstandard mechanism. Namely, it is not driven by the nonlinearity due to the *inertial term* in balance of linear momentum, but it must be attributed to the *nonlinearity in the governing equation for the “elastic” part of the Cauchy stress tensor*. The key challenge is to identify the parameter values that prohibit the onset of “elastic” instability or that trigger the “elastic” instability, and to describe the transition scenarios leading from the laminar to the “turbulent” flow. This task requires one to perform some sort of nonlinear stability analysis, since a nonlinear interaction between the finite amplitude perturbations might be decisive.

The phenomenon of “elastic turbulence” has been thoroughly investigated both from the experimental as well as theoretical point of view (see reviews by Petrie and Denn [1], Larson [2], Shaqfeh [3], Morozov and van Saarloos [4] or Li et al. [5]). In particular, the experiments reported by Groisman and Steinberg [6] stimulated enormous research activity regarding the elastic turbulence. On the other hand, theoretical results mainly follow from direct numerical simulations based on various viscoelastic rate-type models (see Dubief et al. [7], Lieu et al. [8], Grilli et al. [9], Page and Zaki [10], Biancofiore et al. [11], Valente et al. [12], Lee and Zaki [13] and Plan et al. [14] for some recent contributions). The need to resort to sophisticated numerical simulations in order to get qualitative insight into the flow dynamics is not surprising.

The reason is that the instabilities in viscoelastic fluids are very likely of *subcritical* nature (see Meulenbroek et al. [15]). The *subcritical* nature of the instability implies, as remarked by Morozov and van Saarloos [4], that
[Linear stability] (if it exists) is not very relevant for the existence of dynamics of the patterns that typically arise before the instability of the base state occurs.
(Note that a similar issue arises even for the standard Navier–Stokes fluid (see, for example, Baggett and Trefethen [16] for a discussion of several low-dimensional models of subcritical transition).) This means that the linear stability analysis, that is stability analysis with respect to *infinitesimal* perturbations, is of limited applicability in the investigation of the transition scenarios, albeit it can still provide important insight into the problem. (See, for example, Beris et al. [17], Blonce [18], Grillet et al. [19] and Pourjafar and Sadeghy [20], Pourjafar and Sadeghy [21] for linear stability analysis of flows of viscoelastic fluids described by the Giesekus viscoelastic rate-type model.) Moreover, quoting again Morozov and van Saarloos [4]
[Subcritical instability] is governed by all kinds of nonlinear self-enhancing interactions and so there is almost never a simple approximation scheme that allows one to explore the infinite dimensional space of interactions in all details, and determine which direction corresponds to the smallest threshold [for instability]. Thus, in practice, one can explore such situations, in theoretical studies as well as in experiments, only for a given class of perturbations.
On top of that, even if the technique such as weakly nonlinear analysis is apparently successful, then, as Meulenbroek et al. [15] put it,
One should also keep in mind that our expansion is only carried out to lowest order in the nonlinearity, so one may wonder about the robustness of these results as long as higher order terms in the expansion are unknown.


In what follows, we want to address the lack of analytical results for the stability problem of flows of viscoelastic fluids subject to *finite amplitude perturbations*. In particular, using a Lyapunov type technique, we investigate the stability of internal *steady* flows of viscoelastic fluids described by the Giesekus model, and *we derive bounds on the values of the Reynolds number and the Weissenberg number that guarantee the flow stability subject to any (finite) perturbation*. The result provides a *sufficient condition* for stability, hence it can be seen as complementary result to the search for the smallest threshold for instability via approximation methods. The derived bounds are interesting not only on their own. What is perhaps equally interesting is the way the bounds are derived. The *derivation heavily relies on the underlying thermodynamical arguments* and the notion of energy, which is an approach that seems to be discouraged in the nonlinear stability analysis of viscoelastic fluids (see Doering et al. [22]).

The paper is organised as follows. In Section 2, we describe the Giesekus model, and we briefly comment on its thermodynamical underpinnings. In particular, we identify the *energy storage* mechanisms and the *entropy production* mechanisms that are implied by the evolution equations for the Giesekus model. Once the thermodynamical background is summarised, we formulate the governing equations for an internal steady flow (see Section 3), and we proceed with the stability analysis of this non-equilibrium steady state. The stability is analysed using a Lyapunov type functional $V_{\mathrm{neq}}$ constructed by the thermodynamically based method proposed by Bulíček et al. [23]. The functional used in the stability analysis of a steady flow $\hat{v}$ in a domain Ω is constructed in Section 4, and it is given by the formula
(1)$$V_{\mathrm{neq}}\left(\tilde{W}∥\hat{W}\right)=\int_{\Omega }\frac{1}{2}{\left|\tilde{v}\right|}^{2}\;dv+\int_{\Omega }\frac{\Xi }{2}\left[-\ln\det\left(I+{\hat{B_{\kappa_{p(t)}}}}^{-1}\tilde{B_{\kappa_{p(t)}}}\right)+\mathrm{Tr}\left({\hat{B_{\kappa_{p(t)}}}}^{-1}\tilde{B_{\kappa_{p(t)}}}\right)\right]\;dv,$$
where $\tilde{v}$ denotes the perturbation of the velocity field, $\tilde{B_{\kappa_{p(t)}}}$ is related to the perturbation of the stress field, and $\hat{B_{\kappa_{p(t)}}}$ is related to the stress field in the steady flow. (See the corresponding sections for the notation.) The fact that Equation (1) can serve as a Lyapunov type functional is closely related to the proper choice of the *distance function* that characterises the proximity of the perturbation W and the corresponding steady state $\hat{W}$. In our case, the distance function is introduced using the Bures–Wasserstein distance
(2)$${\mathrm{dist}}_{P(d),\;\mathrm{BW}}(A,B){=}_{\mathrm{def}}{\left\{\mathrm{Tr}A+\mathrm{Tr}B-2\mathrm{Tr}\left[{\left(A^{\frac{1}{2}}BA^{\frac{1}{2}}\right)}^{\frac{1}{2}}\right]\right\}}^{\frac{1}{2}},$$
(see Bhatia et al. [24]), which measures the distance between the symmetric positive semidefinite matrices A and B. Once we generalise Equation (2) to the setting of spatially distributed fields of symmetric positive semidefinite matrices, we exploit the concept of Lyapunov functional, and we derive bounds on the Reynolds number and the Weissenberg number that guarantee the flow stability with respect to *any* perturbation (see Theorem 1). These bounds are universal for any flow geometry.

The bounds on the Reynolds number and the Weissenberg number are then explicitly evaluated in Section 6 in the case of Taylor–Couette type flow. Further, we also perform direct numerical simulations that allow us to quantitatively document some features of the perturbation dynamics. The results are commented in Section 7.

## 2. Giesekus Model

### 2.1. Governing Equations

The governing equations for the Giesekus fluid (see Giesekus [25]), in the absence of external force, read
(3a)$$\begin{array}{cc} \mathrm{div}v & =0, \end{array}$$
(3b)$$\begin{array}{cc} \rho \frac{dv}{dt} & =\mathrm{div}T, \end{array}$$
(3c)$$\begin{array}{cc} \nu_{1}\overset{▿}{\bar{B_{\kappa_{p(t)}}}} & =-\mu \left[\alpha B_{\kappa_{p(t)}}^{2}+\left(1-2\alpha \right)B_{\kappa_{p(t)}}-\left(1-\alpha \right)I\right], \end{array}$$
where v denotes the velocity, ρ denotes the density and $B_{\kappa_{p(t)}}$ is an extra tensorial quantity whose physical meaning is given below. Finally, the symbol T denotes the Cauchy stress tensor that is given by the formulae
(3d)$$T=mI+T_{\delta },\;T_{\delta }=2\nu D+\mu {\left(B_{\kappa_{p(t)}}\right)}_{\delta },$$
where *m* denotes the mean normal stress (pressure) and $D{=}_{\mathrm{def}}\frac{1}{2}\left(\nabla v+{\left(\nabla v\right)}^{⊤}\right)$ denotes the symmetric part of the velocity gradient. Symbols ν, $\nu_{1}$, μ and α, α∈(0,1) denote material parameters. Note that, if α=0, then one recovers the standard Maxwell/Oldroyd-B models. *The value*
α=0
*is however not covered in the presented stability analysis*.

The remaining notation is the standard one: $\frac{d}{dt}{=}_{\mathrm{def}}\frac{\partial }{\partial t}+v•\nabla$ denotes the material time derivative, and
(4)$$\overset{▿}{A}{=}_{\mathrm{def}}\frac{dA}{dt}-LA-AL^{⊤},$$
denotes the upper convected derivative, where $L{=}_{\mathrm{def}}\nabla v$, and the symbol $A_{\delta }{=}_{\mathrm{def}}A-\frac{1}{3}\left(\mathrm{Tr}A\right)I$ denotes the traceless part of the corresponding tensor. In virtue of the incompressibility constraint in Equation (3a), we have $D_{\delta }=D$. Note that, if one uses a simple substitution $S{=}_{\mathrm{def}}\mu \left(B_{\kappa_{p(t)}}-I\right)$, and if one redefines the pressure, $p{=}_{\mathrm{def}}-m+\frac{1}{3}\left(\mathrm{Tr}S\right)I$, then Equation (3c) and (3d) transform to $\lambda \overset{▿}{\bar{S}}+S+\frac{\alpha \lambda }{\nu_{1}}S^{2}=2\nu_{1}D$ and T=−pI+2νD+S, where $\lambda {=}_{\mathrm{def}}\frac{\nu_{1}}{\mu }$. This is another frequently used form of the governing equations for the Giesekus fluid.

### 2.2. Thermodynamic Basis

The Giesekus model has been originally derived without any reference to thermodynamics. However, we want to design a Lyapunov type functional using concepts from non-equilibrium thermodynamics, hence we need to explore thermodynamical underpinnings of the model. The issue of finding a thermodynamic basis for viscoelastic rate-type models is claimed to be resolved by a plethora of theories for thermodynamics of complex fluids (see, for example, Leonov [26], Mattos [27], Wapperom and Hulsen [28], Dressler et al. [29] or Ellero et al. [30]). (Notably, the treatise by Dressler et al. [29] contains a rich bibliography, and describes the issue from the viewpoint of the GENERIC formalism (see Grmela and Öttinger [31], Öttinger and Grmela [32] and Pavelka et al. [33])). In the present analysis, we exploit the approach proposed by Rajagopal and Srinivasa [34] that is relatively simple and that provides one a purely phenomenologically based concept of *visco*-*elastic* response.

The fact that the approach by Rajagopal and Srinivasa [34] closely follows the phenomenological concept of *visco*-*elastic* material is best seen in the interpretation of the quantity $B_{\kappa_{p(t)}}$ that appears in the formula for the Cauchy stress tensor. This quantity can be interpreted as the left Cauchy–Green tensor associated with the *elastic* part of the fluid response. Using the approach by Rajagopal and Srinivasa [34], the derivation of Maxwell/Oldroyd-B type models was discussed by Málek et al. [35] (see also Hron et al. [36]). More complex viscoelastic rate-type models that document the applicability of the approach in more involved settings are discussed in Málek et al. [37], Málek et al. [38] and in Dostalík et al. [39]. Following Dostalík et al. [39], we know that the Giesekus fluid is a fluid with the specific Helmholtz free energy ψ in the form
(5)$$\psi {=}_{\mathrm{def}}-c_{V}^{\mathrm{iNSE}}\theta \left[\ln\left(\frac{\theta }{\theta_{\mathrm{ref}}}\right)-1\right]+\frac{\mu }{2\rho }\left(\mathrm{Tr}B_{\kappa_{p(t)}}-3-\ln\det B_{\kappa_{p(t)}}\right),$$
where θ denotes the absolute temperature, $\theta_{\mathrm{ref}}$ denotes a constant reference temperature, $c_{V}^{\mathrm{iNSE}}$ is a positive material parameter (specific heat capacity at constant volume) and μ is another positive material parameter. The specific Helmholtz free energy describes the *energy storage ability* of the fluid, and the chosen ansatz is the same as for the standard Maxwell/Oldroyd-B fluid. This implies that the Giesekus fluid and Maxwell/Oldroyd-B fluids differ, from the perspective of the current approach, only in their *entropy production* mechanisms (see below).

Specifying the Helmholtz free energy as a function of θ and $B_{\kappa_{p(t)}}$, one can use the standard thermodynamical identities $\eta =-\frac{\partial \psi }{\partial \theta }$ and e=ψ+θη, and obtain explicit formulae for the specific entropy η, and the specific internal energy *e*
(6a)$$\eta =c_{V}^{\mathrm{iNSE}}\ln\left(\frac{\theta }{\theta_{\mathrm{ref}}}\right),\;e=c_{V}^{\mathrm{iNSE}}\theta +\frac{\mu }{2\rho }\left(\mathrm{Tr}B_{\kappa_{p(t)}}-3-\ln\det B_{\kappa_{p(t)}}\right).$$

Note that adding the kinetic energy to the mechanical part of the internal energy *e*, that is to the term $\frac{\mu }{2\rho }\left(\mathrm{Tr}B_{\kappa_{p(t)}}-3-\ln\det B_{\kappa_{p(t)}}\right)$, we can define the specific *mechanical energy* via
(6b)$$e_{\mathrm{mech}}{=}_{\mathrm{def}}\frac{1}{2}{\left|v\right|}^{2}+\frac{\mu }{2\rho }\left(\mathrm{Tr}B_{\kappa_{p(t)}}-3-\ln\det B_{\kappa_{p(t)}}\right).$$

Once the Helmholtz free energy, and consequently also the internal energy, is specified, one can derive the evolution equation for the entropy that has the structure $\rho \frac{d\eta }{dt}+\mathrm{div}j_{\eta }=\xi ,$ where $j_{\eta }$ denotes the entropy flux and ξ stands for the entropy production. In the case of Giesekus fluid the entropy production is given by the formula $\xi =\frac{\zeta }{\theta }$, where
(7)$$\zeta {=}_{\mathrm{def}}2\nu D:D+\frac{\mu^{2}}{2\nu_{1}}\mathrm{Tr}\left[\alpha B_{\kappa_{p(t)}}^{2}+\left(1-3\alpha \right)B_{\kappa_{p(t)}}+\left(1-\alpha \right)B_{\kappa_{p(t)}}^{-1}+\left(3\alpha -2\right)I\right]+\kappa \frac{{\left|\nabla \theta \right|}^{2}}{\theta }.$$
(We use the notation $A:A{=}_{\mathrm{def}}\mathrm{Tr}\left(AA^{⊤}\right)$ for the scalar product on the space of matrices, and |A| for the corresponding Frobenius norm.) Since $B_{\kappa_{p(t)}}$ is a symmetric positive definite matrix, it is easy to check that the entropy production in Equation (7) is a nonegative quantity, hence the second law of thermodynamics is satisfied.

The fact that $B_{\kappa_{p(t)}}$ is a symmetric positive definite matrix follows directly from the governing equations in Equation (3) via an argument similar to that of Boyaval et al. [40]. It is also a consequence of the fact that $B_{\kappa_{p(t)}}$ is in the approach by Rajagopal and Srinivasa [34] constructed as the left Cauchy–Green tensor, which means that $B_{\kappa_{p(t)}}$ can be decomposed as $B_{\kappa_{p(t)}}=F_{\kappa_{p(t)}}F_{\kappa_{p(t)}}^{⊤}$. We exploit the positivity of $B_{\kappa_{p(t)}}$ quite frequently in our analysis.

Finally, we introduce three more important quantities that play crucial role in the construction of Lyapunov type functional via the method proposed by Bulíček et al. [23]. Namely, we introduce the *net total energy*
$E_{\mathrm{tot}}$, the *net mechanical energy*
$E_{\mathrm{mech}}$ and the *net entropy S* of the fluid occupying the domain Ω,
(8)$$E_{\mathrm{tot}}{=}_{\mathrm{def}}\int_{\Omega }\rho \left[\frac{1}{2}{\left|v\right|}^{2}+e\right]\;dv,\;E_{\mathrm{mech}}{=}_{\mathrm{def}}\int_{\Omega }\rho e_{\mathrm{mech}}\;dv,\;S{=}_{\mathrm{def}}\int_{\Omega }\rho \eta \;dv.$$

### 2.3. Scaling

The equations in Equation (3) governing the evolution of mechanical variables can be transformed to a dimensionless form by introducing the characteristic length $x_{\mathrm{char}}$, characteristic time $t_{\mathrm{char}}$. (Note that the tensor field $B_{\kappa_{p(t)}}$ already is a dimensionless quantity.) Using the following relations between the original quantities and their dimensionless versions denoted by stars $x=x_{\mathrm{char}}x^{⋆}$, $t=t_{\mathrm{char}}t^{⋆}$, $v=\frac{x_{\mathrm{char}}}{t_{\mathrm{char}}}v^{⋆}$, $m=\frac{\nu }{t_{\mathrm{char}}}m^{⋆}$, we obtain
(9a)$$\begin{array}{cc} {\mathrm{div}}^{⋆}v^{⋆} & =0, \end{array}$$
(9b)$$\begin{array}{cc} \frac{dv^{⋆}}{dt^{⋆}} & ={\mathrm{div}}^{⋆}T^{⋆}, \end{array}$$
(9c)$$\begin{array}{cc} {\overset{▿}{\bar{B_{\kappa_{p(t)}}}}}^{⋆} & =-\frac{1}{\mathrm{Wi}}\left[\alpha B_{\kappa_{p(t)}}^{2}+\left(1-2\alpha \right)B_{\kappa_{p(t)}}-\left(1-\alpha \right)I\right], \end{array}$$
where the dimensionless Cauchy stress tensor $T^{⋆}$ is given by
(9d)T⋆=1Rem⋆I+(T⋆)δ,(T⋆)δ=2Re(D⋆)δ+Ξ(Bκp(t))δ.

In Equation (9), we introduce three dimensionless numbers—the *Reynolds number*
Re, the *Weissenberg number*
Wi and dimensionless shear modulus Ξ—via the formulae Re=defρxchar2νtchar, $\mathrm{Wi}{=}_{\mathrm{def}}\frac{\nu_{1}}{\mu t_{\mathrm{char}}}$, and $\Xi {=}_{\mathrm{def}}\frac{\mu t_{\mathrm{char}}^{2}}{\rho x_{\mathrm{char}}^{2}}$. It remains to introduce a scaling factor for the net mechanical energy $E_{\mathrm{mech}}$, which is used for the construction of the Lyapunov type functional in Section 4. Using the scaling $E_{\mathrm{mech}}=\frac{\rho x_{\mathrm{char}}^{5}}{t_{\mathrm{char}}^{2}}E_{\mathrm{mech}}^{⋆}$, we obtain
(10)$$E_{\mathrm{mech}}^{⋆}\left(W^{⋆}\right)=\int_{\Omega^{⋆}}\left[\frac{1}{2}{\left|v^{⋆}\right|}^{2}+\frac{\Xi }{2}\left(\mathrm{Tr}B_{\kappa_{p(t)}}-3-\ln\det B_{\kappa_{p(t)}}\right)\right]\;dv^{⋆}.$$

Hereafter, we omit the star denoting dimensionless quantities unless otherwise specified.

The scaling is chosen in such a way that, if Wi→0+, then $B_{\kappa_{p(t)}}$ approaches the identity tensor. Indeed, if Wi→0+ then, Equation (9c) implies that
(11)$$O=\alpha B_{\kappa_{p(t)}}^{2}+\left(1-2\alpha \right)B_{\kappa_{p(t)}}-\left(1-\alpha \right)I,$$
and the solution of Equation (11) is $B_{\kappa_{p(t)}}=I$. Moreover, if $B_{\kappa_{p(t)}}=I$, then the second term in Equation (10), that is the elastic contribution to the mechanical energy, vanishes, and the mechanical energy of the fluid reduces to the kinetic energy only. Finally, if $B_{\kappa_{p(t)}}=I$, then the additional term in the Cauchy stress tensor in Equation (9d) vanishes. This means that for Wi→0+ the governing equations in Equation (9) reduce to the standard incompressible Navier–Stokes equations.

### 2.4. Boundary Conditions

The governing equations in Equation (9) must be supplemented with boundary conditions for the velocity v. We are interested in *internal flow* problems, where one prescribes Dirichlet boundary conditions on a part of the flow domain $\Omega \subset R^{3}$, and periodic boundary conditions on another part of the domain. Such a domain is usually called the periodic cell. (For example, in the case of flow in between two infinite concentric rotating cylinders, the Dirichlet boundary condition is prescribed on the surfaces of the cylinders, while the periodic boundary condition is prescribed in the direction of the axis of the cylinders.) On the parts of the boundary corresponding to the periodicity directions, say $\Gamma_{1}$, we therefore prescribe periodic boundary condition for v, while, on the remaining part of the boundary, say $\Gamma_{2}$, we prescribe the no-penetration and the no-slip boundary condition,
(12a)$$\begin{array}{cc} {v•n|}_{\Gamma_{2}} & =0, \end{array}$$
(12b)$$\begin{array}{cc} {\left(I-n⊗n)v\right|}_{\Gamma_{2}} & =V, \end{array}$$
where n is the unit outward normal to the boundary of Ω and V is a given velocity in the tangential direction to the boundary. This means that the fluid adheres to the boundary, and, moreover, if V≠0, then, in general, *the energy is exchanged between the fluid and its surroundings*. Indeed, the balance of the net total energy reads $\frac{dE_{\mathrm{tot}}}{dt}=\int_{\partial \Omega }\left(Tv\right)•n\;ds-\int_{\partial \Omega }j_{q}•n\;ds,$ where ∂Ω denotes the boundary of the domain Ω and $j_{q}$ denotes the heat flux. Consequently, if v≠0 on the boundary, then the term $\int_{\partial \Omega }\left(Tv\right)•n\;ds$ does not, in general, vanish or is compensated by the second term on the right-hand side, and the *net total energy* might even change in time.

Concerning the boundary conditions for the perturbation $\tilde{v}$ with respect to the reference state $\hat{v}$ (see below), we see that, if $\hat{v}$ satisfies Equation (12), then the perturbed state $v=\hat{v}+\tilde{v}$ also satisfies Equation (12) provided that
(13)$${\tilde{v}|}_{\Gamma_{2}}=0.$$
The periodic boundary condition on $\Gamma_{1}$ is preserved for the perturbation $\tilde{v}$.

In the following, we frequently use the identity
(14)$$\int_{\partial \Omega }f•n\;ds=0,$$
where $f:\partial \Omega \to R^{3}$ is a smooth function such that f fulfills the periodic boundary condition on $\Gamma_{1}$ and f=0 on $\Gamma_{2}$. Note that the identity holds even if one part of the boundary, whether $\Gamma_{1}$ or $\Gamma_{2}$, is not present.

## 3. Base Flow—Non-Equilibrium Steady State

### 3.1. Notation for the Stability Analysis

We are interested in the evolution of the triplet $W{=}_{\mathrm{def}}\left[v,m,B_{\kappa_{p(t)}}\right],$ which solves the evolution equations in Equation (3). We further use the notation $\hat{W}=\left[\hat{v},\hat{m},\hat{B_{\kappa_{p(t)}}}\right]$ for the triplet corresponding to a *non-equilibrium steady state solution*, and $\tilde{W}=\left[\tilde{v},\tilde{m},\tilde{B_{\kappa_{p(t)}}}\right]$ for the *perturbation with respect to the non-equilibrium steady state*. This means that the triplet describing the complete perturbed state is given as a sum of the reference state $\hat{W}$ and the perturbation $\tilde{W}$ with respect to the reference state
(15a)$$\begin{array}{cc} W & =\hat{W}+\tilde{W}, \end{array}$$
(15b)$$\begin{array}{cc} \left[v,m,B_{\kappa_{p(t)}}\right] & =\left[\hat{v},\hat{m},\hat{B_{\kappa_{p(t)}}}\right]+\left[\tilde{v},\tilde{m},\tilde{B_{\kappa_{p(t)}}}\right]. \end{array}$$
(Note that sometimes we work only with the pair $W{=}_{\mathrm{def}}\left[v,B_{\kappa_{p(t)}}\right]$, since the pressure is insubstantial in our analysis.) The term *non-equilibrium steady state* is chosen in accordance with the practice in thermodynamics, and it means that the *entropy is produced* at the steady state $\hat{W}$. In particular, the adjective non-equilibrium does not refer to the stability of the steady state.

### 3.2. Governing Equations in a Steady State

*The steady state*$\hat{W}=\left[\hat{v},\hat{m},\hat{B_{\kappa_{p(t)}}}\right]$*whose stability we want to investigate is a solution to the equations in Equation (9) where the partial time derivatives are identically equal to zero.* In particular, we assume that the state described by the triplet $\left[\hat{v},\hat{m},\hat{B_{\kappa_{p(t)}}}\right]$ solves the system
(16a)$$\begin{array}{cc} \mathrm{div}\hat{v} & =0, \end{array}$$
(16b)$$\begin{array}{cc} \left(\hat{v}•\nabla \right)\hat{v} & =\mathrm{div}T(\hat{W}), \end{array}$$
(16c)$$\begin{array}{cc} \left(\hat{v}•\nabla \right)\hat{B_{\kappa_{p(t)}}}-\hat{L}\hat{B_{\kappa_{p(t)}}}-\hat{B_{\kappa_{p(t)}}}{\hat{L}}^{⊤} & =-\frac{1}{\mathrm{Wi}}\left[\alpha {\hat{B_{\kappa_{p(t)}}}}^{2}+\left(1-2\alpha \right)\hat{B_{\kappa_{p(t)}}}-\left(1-\alpha \right)I\right]. \end{array}$$
subject to the boundary conditions in Equation (12) on $\Gamma_{2}$, that is
(17)$${\hat{v}•n|}_{\Gamma_{2}}=0,\;{\left(I-n⊗n\right)\hat{v}|}_{\Gamma_{2}}=V,$$
and the periodic boundary conditions on $\Gamma_{1}$. Here, the symbol $T(\hat{W})$ denotes the Cauchy stress tensor induced by the triplet $\left[\hat{v},\hat{m},\hat{B_{\kappa_{p(t)}}}\right]$, that is
(18)T(W^)=1Rem^I+2ReD^+Ξ(Bκp(t)^)δ,
where $\hat{D}=\frac{1}{2}\left(\hat{L}+{\hat{L}}^{⊤}\right)$, and $\hat{L}=\nabla \hat{v}$.

Note that, if V=0, that is if no mechanical energy is supplied to the fluid, then the system would admit an *equilibrium solution*
(19)$$\left[\hat{v},\hat{m},\hat{B_{\kappa_{p(t)}}}\right]=\left[0,c,I\right],$$
where *c* is an arbitrary number. (This is the standard ambiguity in the identification of the pressure well known from the case of Navier–Stokes fluid.) Here, we use the adjective *equilibrium* to emphasise that such a steady state would lead to *zero entropy production*. Indeed, if $B_{\kappa_{p(t)}}=I$ and v=0, then the (mechanical part) of the entropy production in Equation (7) vanishes.

On the other hand, if V≠0, then one must in general expect that the steady fields $\hat{v}$ and $\hat{B_{\kappa_{p(t)}}}$ are *spatially inhomogeneous*, and consequently the entropy production in Equation (7) is positive. This means that the system *produces the entropy*; hence, it is, from the thermodynamical point of view, *out of equilibrium*. Consequently, as discussed above, we use the adjective *non-equilibrium* and we refer to the base flow as of non-equilibrium steady state.

### 3.3. Concept of Stability

Concerning the stability of the non-equilibrium steady state, we are interested in its *asymptotic stability*. If we have a non-equilibrium steady state $\hat{W}$ that solves Equation (16), then we want to know whether the perturbation $W=\hat{W}+\tilde{W}$ of the non-equilibrium steady state $\hat{W}$ tends back to the non-equilibrium steady state $\hat{W}$ as time goes to infinity. In our case, the evolution of the perturbed state W is governed by the equations in Equation (9) that must be solved subject to the given boundary conditions in Equation (12) and subject to initial conditions
(20)$${v|}_{t=0}=\hat{v}+{\tilde{v}}_{0},\;{B_{\kappa_{p(t)}}|}_{t=0}=\hat{B_{\kappa_{p(t)}}}+{\left(\tilde{B_{\kappa_{p(t)}}}\right)}_{0}.$$

The non-equilibrium steady state $\hat{W}$ is said to be *asymptotically stable* provided that the solutions that start close enough to the steady state not only remain close enough to the steady state but also eventually converge to the steady state, that is the triplet W converges to $\hat{W}$ as time goes to infinity,
(21)$$W\overset{t\to +\infty }{\to }\hat{W},$$
for all sufficiently small initial data ${\tilde{v}}_{0}$ and ${\left(\tilde{B_{\kappa_{p(t)}}}\right)}_{0}$ (see (Henry [41], Definition 4.1.2) or Yoshizawa [42] for the formal definition and further discussion).

Ideally, one would like to obtain stronger results. Namely, one would like to have an *unconditional* result that states that the non-equilibrium steady state is recovered as time goes to infinity *regardless of the choice of initial perturbation*. This behaviour is expected if one deals with non-equilibrium steady states that are driven by a small energy inflow that is by a small boundary velocity V, or, in other words, if one deals with non-equilibrium steady states that are not far away from the equilibrium steady state.

The key task in the stability analysis is the *choice of a metric/norm on the state space* to give a meaning to the statement in Equation (21). Namely, we need to answer the question as how to characterise the distance between $\hat{W}$ and W, since Equation (21) means
(22)$$\mathrm{dist}\left(\hat{W},W\right)\overset{t\to +\infty }{\to }0,$$
where dist(·,·) is a given metric that is not necessarily induced by a norm. Since $B_{\kappa_{p(t)}}$ is at a given spatial point x∈Ω a positive definite matrix, it seems reasonable to design the metric in such a way that it reflects this fact. This means that we have to rely on a *metric on the set of positive definite matrices*. There are several possible definitions of the metric on these sets (see Appendix A). If we use the Bures–Wasserstein distance,
(23)$${\mathrm{dist}}_{P(d),\;\mathrm{BW}}(A,B){=}_{\mathrm{def}}{\left\{\mathrm{Tr}A+\mathrm{Tr}B-2\mathrm{Tr}\left[{\left(A^{\frac{1}{2}}BA^{\frac{1}{2}}\right)}^{\frac{1}{2}}\right]\right\}}^{\frac{1}{2}},$$
(see Equation (A1) in Appendix A), and if we generalise this concept to the spatially distributed tensor fields, then we can define the distance between $\hat{W}$ and W as
(24)$$\mathrm{dist}\left(\hat{W},W\right){=}_{\mathrm{def}}{\left({∥\hat{v}-v∥}_{L^{2}\left(\Omega \right)}^{2}+{\left[{\mathrm{dist}}_{P_{\Omega }\left(d\right),\;\mathrm{BW}}\left(\hat{B_{\kappa_{p(t)}}},B_{\kappa_{p(t)}}\right)\right]}^{2}\right)}^{\frac{1}{2}},$$
(see Equation (A7) and Appendix A for a discussion of the notation and correctness of this definition). It turns out that this concept of distance nicely fits to the dynamical system we are interested in.

The term “stability” is used in many other contexts; hence, we briefly comment on these other notions of stability. In particular, we emphasise what is in the present work *not* meant by the stability. First, we are not interested in the *stability in the sense of continuous dependence on initial data*, which is the concept of stability investigated in Dafermos [43] and various subsequent works especially in the theory of hyperbolic systems (see Dafermos [44]). The stability in the sense of continuous dependence on initial data means (see, for example, Schaeffer and Cain [45]) that
[…] if the initial data for an initial value problem are altered slightly, then the perturbed solution diverges from the original solution no faster than at a controlled exponential rate.
Apparently, the *asymptotic stability* we are interested in is a more ambitious concept, since we want the perturbed solution to *converge* back to the original solution (non-equilibrium steady state). Second, we are not interested in the *stability of the steady state subject to infinitesimal perturbations*, that is in the linearised stability. We are interested in the evolution of *finite amplitude perturbations*.

Finally, we emphasise that in our analysis *we work with perturbations that are solution to the governing equations in the classical sense*. (All derivatives are understood as the classical derivatives, not as generalised derivatives such as distributional derivatives and so forth.) In particular, we *do not* consider the perturbations that solve the governing equations only in a weak sense, although it is an important issue worth further investigation. The reader interested in the discussion of the state-of-the-art rigorous mathematical theory of equations governing the motion of viscoelastic fluids is kindly referred to the work of Masmoudi [46] or Barrett and Süli [47].

## 4. Lyapunov Functional

### 4.1. Concept of Lyapunov Functional

Let us briefly recall the concept of Lyapunov functional (see Henry [41]). We consider a system of governing equations in the form
(25)$$\frac{dX}{dt}=F\left(X\right),$$
where $\hat{X}$ is a steady state, that is $F(\hat{X})=0$, and where ${∥\cdot ∥}_{\mathrm{st}}$ denotes a norm on the underlying state space. We say that the functional $V(\tilde{X}∥\hat{X})$ is a strict Lyapunov functional of the steady state $\hat{X}$ provided that:
There exists a neighbourhood of $\hat{X}$ such that the functional is bounded from below by a function *f* of the distance between the steady state $\hat{X}$ and the perturbation X, that is
(26a)$$V\left(\tilde{X}∥\hat{X}\right)\geq f\left({∥\hat{X}-X∥}_{\mathrm{st}}\right),$$
where *f* is a continuous strictly increasing function such that f(0)=0 and f(r)>0 whenever r>0.The time derivative of $V(\tilde{X}∥\hat{X})$ is negative and bounded from above by a function *g* of the distance between the steady state $\hat{X}$ and the perturbation X, that is
(26b)$$\frac{d}{dt}V\left(\tilde{X}∥\hat{X}\right)\leq -g\left({∥\hat{X}-X∥}_{\mathrm{st}}\right),$$
where *g* is a continuous strictly increasing function such that g(0)=0 and g(r)>0 whenever r>0.

If the given system of governing equations admits a strict Lyapunov functional near the state $\hat{X}$, then we know that the steady state $\hat{X}$ is asymptotically stable (see (Henry [41], Theorem 4.1.4)). This means that the solution $X=\tilde{X}+\hat{X}$ that starts in the neighbourhood of $\hat{X}$ satisfies
(27)$${∥\hat{X}-X∥}_{\mathrm{st}}\overset{t\to +\infty }{\to }0.$$

While the concept of Lyapunov type functional is very simple, it is difficult to apply in a particular setting. The main difficulty is to find a Lyapunov type functional. (Note that in the infinite dimensional setting one can not easily exploit LaSalle’s invariance principle since it requires precompactness of the trajectories, which is a qualitative property that goes beyond our assumption regarding the existence of the classical solution. Consequently, having a Lyapunov type functional with Equation (26b) replaced by the mere negativity everywhere except at the equilibrium, $\frac{d}{dt}V\left(\tilde{X}∥\hat{X}\right)<0$ for $\tilde{X}\neq 0$, is not a viable option.) Fortunately, since we are interested in equations describing a physical system, we can try to search for the functional using physical concepts.

If we were dealing with the stability of *a homogeneous equilibrium steady state in a thermodynamically isolated system*, then a Lyapunov type functional could be constructed using the net entropy *S* and the net total energy $E_{\mathrm{tot}}$ functional. It can be shown that the appropriate Lyapunov type functional is in this setting reads
(28)$$V_{\mathrm{eq}}\left(\tilde{W}∥\hat{W}\right)=-\left[S\left(\hat{W}+\tilde{W}\right)-\frac{1}{\hat{\theta }}\left\{E_{\mathrm{tot}}\left(\hat{W}+\tilde{W}\right)-E_{\mathrm{tot}}\left(\hat{W}\right)\right\}\right],$$
where $\hat{\theta }$ is the *spatially homogeneous temperature in the equilibrium steady state* (see Bulíček et al. [23] for details). The observation that Equation (28) can be used as a suitable Lyapunov type functional for stability analysis of *homogeneous equilibrium steady states* in a thermodynamically *isolated systems* or *mechanically isolated systems immersed in a thermal bath* is well known (see, for example, Šilhavý [48] or Grmela and Öttinger [31], Öttinger and Grmela [32]), and in the continuum thermodynamics setting it dates back to the works of Coleman [49], Gurtin [50], and Gurtin [51]. (The core idea can be found in earlier works, see especially Duhem [52].) Unfortunately, the same functional cannot be used in stability analysis of *non-equilibrium spatially inhomogeneous steady states* in thermodynamically *open* systems. This fact is clear from Equation (28) itself. If one works with a *spatially inhomogeneous* steady states, then $\hat{\theta }$ in Equation (28) is a *function*, and Equation (28) does not define a functional at all.

### 4.2. Construction of Lyapunov Type Functional for Stability Analysis of a Spatially Inhomogeneous Steady State

Recently, Bulíček et al. [23] proposed a method for construction of Lyapunov type functionals for stability analysis of non-equilibrium spatially *inhomogeneous steady states* in thermodynamically *open* systems. In the ongoing analysis, we use the same ideas as in Bulíček et al. [23], but we restrict ourselves to the *mechanical quantities only*. This is a matter of convenience, since we are interested in mechanical quantities only, and the temperature evolution has no feedback on the mechanical part of the system of governing equations. Consequently, we do not need to work with the Lyapunov type functional for the full thermomechanical problem, and we can construct a simpler Lyapunov type functional solely for the mechanical quantities.

Using the net mechanical energy functional $E_{\mathrm{mech}}$ introduced in Equation (10), one can see that the net mechanical energy in a thermodynamically closed system must decay in time. Consequently, the functional
(29)$$V_{\mathrm{eq}}\left(\tilde{W}∥\hat{W}\right){=}_{\mathrm{def}}E_{\mathrm{mech}}\left(\hat{W}+\tilde{W}\right)-E_{\mathrm{mech}}\left(\hat{W}\right)$$
can serve as a Lyapunov type functional for stability analysis of equilibrium spatially homogeneous state in Equation (19).

Following the methodology outlined in Bulíček et al. [23], we use the Lyapunov type functional for the *equilibrium* steady state in Equation (29), and we define the candidate for the Lyapunov type functional for the *non-equilibrium* steady state as
(30)$$V_{\mathrm{neq}}\left(\tilde{W}∥\hat{W}\right){=}_{\mathrm{def}}E_{\mathrm{mech}}\left(\hat{W}+\tilde{W}\right)-E_{\mathrm{mech}}\left(\hat{W}\right)-{D_{W}E_{\mathrm{mech}}\left(W\right)|}_{W=\hat{W}}\left[\tilde{W}\right],$$
where ${D_{W}E_{\mathrm{mech}}\left(W\right)|}_{W=\hat{W}}\left[\tilde{W}\right]$ denotes the Gâteaux derivative at point $\hat{W}$ in the direction $\tilde{W}$. (This is essentially the affine correction trick introduced in a different context by Ericksen [53] (see also Dafermos [43]).) The (dimensionless) explicit formulae for the individual terms in Equation (30) read
(31a)$$\begin{array}{cc} E_{\mathrm{mech}}\left(\hat{W}+\tilde{W}\right) & =\int_{\Omega }\left[\frac{1}{2}{\left|\hat{v}+\tilde{v}\right|}^{2}+\frac{\Xi }{2}\left(\mathrm{Tr}\left(\hat{B_{\kappa_{p(t)}}}+\tilde{B_{\kappa_{p(t)}}}\right)-3-\ln\det\left(\hat{B_{\kappa_{p(t)}}}+\tilde{B_{\kappa_{p(t)}}}\right)\right)\right]\;dv, \end{array}$$
(31b)$$\begin{array}{cc} E_{\mathrm{mech}}\left(\hat{W}\right) & =\int_{\Omega }\left[\frac{1}{2}{\left|\hat{v}\right|}^{2}+\frac{\Xi }{2}\left(\mathrm{Tr}\hat{B_{\kappa_{p(t)}}}-3-\ln\det\hat{B_{\kappa_{p(t)}}}\right)\right]\;dv, \end{array}$$
(31c)$$\begin{array}{cc} {D_{W}E_{\mathrm{mech}}\left(W\right)|}_{W=\hat{W}}\left[\tilde{W}\right] & =\int_{\Omega }\left\{\hat{v}•\tilde{v}+\frac{\Xi }{2}\left[\mathrm{Tr}\tilde{B_{\kappa_{p(t)}}}-\mathrm{Tr}\left({\hat{B_{\kappa_{p(t)}}}}^{-1}\tilde{B_{\kappa_{p(t)}}}\right)\right]\right\}\;dv. \end{array}$$

Using Equation (31) in Equation (30), we get, after some algebraic manipulations, the explicit formula for the proposed Lyapunov type functional
(32)$$V_{\mathrm{neq}}\left(\tilde{W}∥\hat{W}\right)=\int_{\Omega }\frac{1}{2}{\left|\tilde{v}\right|}^{2}\;dv+\int_{\Omega }\frac{\Xi }{2}\left[-\ln\det\left(I+{\hat{B_{\kappa_{p(t)}}}}^{-1}\tilde{B_{\kappa_{p(t)}}}\right)+\mathrm{Tr}\left({\hat{B_{\kappa_{p(t)}}}}^{-1}\tilde{B_{\kappa_{p(t)}}}\right)\right]\;dv.$$

It remains to show that the functional in Equation (32) has the properties in Equation (26) introduced in Section 4.1. First, we show that the condition in Equation (26a) holds for a neighbourhood of $\hat{W}$, which means that we have to specify a norm on the state space.

The suitable norm is the norm introduced in Appendix A in Definition A3. This norm is a “standard” Lebesgue type norm. However, it turns out that it is convenient to use this norm for a characterisation of the evolution of a “shifted” state. The idea is the following. If we are given a *constant-in-time* tensor field $\hat{B_{\kappa_{p(t)}}}$, which corresponds to the steady solution of Equation (9), and a state $W=[v,B_{\kappa_{p(t)}}]$, then we can introduce the shifted state $Z=[v,Z_{\kappa_{p(t)}}]$ that is defined as
(33)$$W=\left[v,B_{\kappa_{p(t)}}\right]\mapsto Z=\left[v,Z_{\kappa_{p(t)}}\right],\;Z_{\kappa_{p(t)}}{=}_{\mathrm{def}}{\left({\hat{B_{\kappa_{p(t)}}}}^{-\frac{1}{2}}B_{\kappa_{p(t)}}{\hat{B_{\kappa_{p(t)}}}}^{-\frac{1}{2}}\right)}^{\frac{1}{2}}.$$

This shifted state seems to be ideal for the investigation of the perturbations to the steady state $\hat{B_{\kappa_{p(t)}}}$. Indeed, the steady state for Equation (9) is $\hat{W}=\left[\hat{v},\hat{B_{\kappa_{p(t)}}}\right]$, which is in virtue of Equation (33) shifted to
(34)$$\hat{Z}=\left[\hat{v},I\right].$$

Now, instead of investigating the behaviour of the perturbation $B_{\kappa_{p(t)}}$ with respect to $\hat{B_{\kappa_{p(t)}}}$, we can investigate the behaviour of the shifted perturbation $Z_{\kappa_{p(t)}}$ with respect to the identity I.

**Lemma** **1**(Relation between the proposed Lyapunov functional and a norm)**.**
*Let*
$\hat{W}$
*and*
$W=\hat{W}+\tilde{W}$
*denote two states of the system governed by equations in Equation (9), and let*
$\hat{Z}$
*and*
Z
*denote the corresponding shifted states. Furthermore, let*
${∥\cdot ∥}_{\mathrm{st}}$
*denote the norm introduced in Definition A3. Then, there exists a positive constant*
D(Ξ)
*such that*
(35)$$V_{\mathrm{neq}}\left(\tilde{W}∥\hat{W}\right)\geq D{∥\hat{Z}-Z∥}_{\mathrm{st}}^{2}.$$

**Proof.** We note that Equation (32) for the Lyapunov type functional can be rewritten as
Vneq(W˜∥W^=12∥v^−(v^+v˜)∥L2(Ω)2+∫ΩΞ2[Tr(Bκp(t)^−12(Bκp(t)^+Bκp(t)˜)Bκp(t)^−12)−3−lndet(Bκp(t)^−12(Bκp(t)^+Bκp(t)˜)Bκp(t)^−12)] dv(36)=12∥v^−(v^+v˜)∥L2(Ω)2+∫ΩΞ2[TrZκp(t)2−3−lndetZκp(t)2] dv,
where we use the cyclic property of the trace and the properties of the determinant. Making use of Lemma A3 and the inequality in the integrand of the last term in Equation (36), we see that
(37)$$V_{\mathrm{neq}}\left(\tilde{W}∥\hat{W}\right)\geq \frac{1}{2}{∥\hat{v}-\left(\hat{v}+\tilde{v}\right)∥}_{L^{2}\left(\Omega \right)}^{2}+\frac{\Xi }{2}\int_{\Omega }{\left|I-Z_{\kappa_{p(t)}}\right|}^{2}\;dv\geq \min\left\{\frac{1}{2},\frac{\Xi }{2}\right\}{∥\hat{Z}-Z∥}_{\mathrm{st}}^{2}.$$  □

The less straightforward part of the analysis of properties of proposed Lyapunov type functional $V_{\mathrm{neq}}$ is the evaluation of its time derivative $\frac{dV_{\mathrm{neq}}}{dt}$. The formula for the time derivative is derived via a lengthy algebraic manipulation described in Appendix B, and it is given below in Lemma 2. Note that, *although we are working with a thermodynamically open system*, the formula for the time derivative *does not contain boundary terms*.

**Lemma** **2**(Explicit formula for the time derivative of the Lyapunov type functional)**.**
*Let*
$\hat{W}$
*and*
W=W^+W˜
*denote two states of the system governed by equations in Equation (9), where the state*
$\hat{W}$
*is the steady state, that is it solves Equation (16). The formula for the time derivative of the functional*
$V_{\mathrm{neq}}\left(\tilde{W}∥\hat{W}\right)$
*introduced in Equation (32) reads*
dVneqdt(W˜∥W^)=−∫Ω2ReD˜:D˜ dv−∫ΩΞ Bκp(t)˜:D˜ dv−∫ΩD^v˜•v˜ dv−∫ΩΞ2Tr[Bκp(t)^−1Bκp(t)˜Bκp(t)^−1(v˜•∇)Bκp(t)^] dv+∫ΩΞ2Bκp(t)^−1:(L˜Bκp(t)˜+Bκp(t)˜L˜⊤) dv−∫Ω(1−α)Ξ2WiTr[(Bκp(t)^+Bκp(t)˜)−1(Bκp(t)˜Bκp(t)^−1)(Bκp(t)˜Bκp(t)^−1)⊤] dv(38)−∫ΩαΞ2WiTr[Bκp(t)^−1Bκp(t)˜2] dv.

**Proof.** See Appendix B.  □

We note that the terms
(39a)−∫Ω2ReD˜:D˜ dv,−∫ΩαΞ2WiTr[Bκp(t)^−1Bκp(t)˜2] dv,
(39b)$$\begin{array}{c} -\int_{\Omega }\left(1-\alpha \right)\frac{\Xi }{2\mathrm{Wi}}\mathrm{Tr}\left[{\left(\hat{B_{\kappa_{p(t)}}}+\tilde{B_{\kappa_{p(t)}}}\right)}^{-1}\left(\tilde{B_{\kappa_{p(t)}}}{\hat{B_{\kappa_{p(t)}}}}^{-1}\right){\left(\tilde{B_{\kappa_{p(t)}}}{\hat{B_{\kappa_{p(t)}}}}^{-1}\right)}^{⊤}\right]\;dv, \end{array}$$
are negative provided that $\tilde{v}\neq 0$ and $\tilde{B_{\kappa_{p(t)}}}\neq O$. If we were able to show that these damping terms are strong enough to balance all the remaining terms on the right-hand side of Equation (38), we would get the desired result (Equation (26b)) concerning the negativity of the time derivative. This should be possible at least for sufficiently small Reynolds number Re and Weissenberg number Wi. The hypothesis follows from the observation that as Re and Wi tend to zero, then the magnitude of the damping terms increases, and it should outgrow the other terms in Equation (38) that do not depend on Re and Wi. This observation is consistent with the expectation that low Reynolds number and low Weissenberg number flows are stable.

Now, the objective is to show that the hypothesis is true, and that the proposed functional indeed satisfies the condition in Equation (26b). In the quantification of the “sufficient smallness” of the Weissenberg number Wi and the Reynolds number Re, we aim at a *simple but very rough estimate* based on the elementary use of Friedrichs–Poincaré, Cauchy–Schwarz, Young and Korn (in)equalities (see Nečas et al. [54] or Evans [55] or any other standard reference work on function spaces). A precise characterisation of the Reynolds number and the Weissenberg number that guarantee the negativity of the time derivative, and hence the stability, could be obtained by a variational technique known from the standard energy method (see Joseph [56] or Straughan [57]). This is however beyond the scope of the current contribution.

**Lemma** **3**(Estimate on the time derivative)**.**
*Let*
$\hat{W}$ and $W=\hat{W}+\tilde{W}$
*denote two states of the system governed by equations in Equation (9), where the state*
$\hat{W}$
*is the steady state, that is it solves Equation (16). Then, there exist constants*
C1(W^,Re,Wi,Ξ,Ω)
*and*
C2(W^,Re,Wi,Ξ)
*such that the time derivative of the functional*
$V_{\mathrm{neq}}\left(\tilde{W}∥\hat{W}\right)$
*introduced in Equation (32) can be estimated from above as*
(40)$$\frac{dV_{\mathrm{neq}}}{dt}\left(\tilde{W}∥\hat{W}\right)\leq C_{1}{∥\nabla \tilde{v}∥}_{L^{2}\left(\Omega \right)}^{2}+C_{2}{∥\tilde{B_{\kappa_{p(t)}}}∥}_{L^{2}\left(\Omega \right)}^{2},$$
*where we denote*
(41a)C1=def−1Re+CPsupx∈Ω|λmin(D^)|+Ξ2supx∈Ω|Bκp(t)^−1−I|+CPΞ4supx∈Ω|Bκp(t)^−1|2supx∈Ω|∇Bκp(t)^|,
(41b)$$C_{2}{=}_{\mathrm{def}}-\alpha \frac{\Xi }{2\mathrm{Wi}}\underset{x\in \Omega }{\inf}\lambda_{\min}\left({\hat{B_{\kappa_{p(t)}}}}^{-1}\right)+\frac{\Xi }{2}\underset{x\in \Omega }{\sup}\left|{\hat{B_{\kappa_{p(t)}}}}^{-1}-I\right|+\frac{\Xi }{4}\underset{x\in \Omega }{\sup}{\left|{\hat{B_{\kappa_{p(t)}}}}^{-1}\right|}^{2}\underset{x\in \Omega }{\sup}\left|\nabla \hat{B_{\kappa_{p(t)}}}\right|,$$
*and where*
$\lambda_{\min}\left(\cdot \right)$
*denotes the minimal eigenvalue of the corresponding matrix and*
$C_{P}$
*denotes the domain dependent constant from Friedrichs–Poincaré inequality.*

**Proof.** See Appendix C.  □

**Lemma** **4**(Estimate on the time derivative in terms of the norm on the shifted state space)**.**
*Let*
$\hat{W}$
*and*
W=W^+W˜
*denote two states of the system governed by equations in Equation (9), where the state*
$\hat{W}$
*is the steady state, that is it solves Equation (16), and let*
$\hat{Z}$ and Z
*denote the corresponding shifted states (see Equation (33)). Let us further assume that the constants*
$C_{1}$
*and*
$C_{2}$
*in Lemma 3 are negative. Then, there exists a positive constant*
$C(C_{1},C_{2},\hat{B_{\kappa_{p(t)}}})$
*such that*
(42)$$\frac{dV_{\mathrm{neq}}}{dt}\left(\tilde{W}∥\hat{W}\right)\leq -C{∥\hat{Z}-Z∥}_{\mathrm{st}}^{2},$$
*where*
$V_{\mathrm{neq}}\left(\tilde{W}∥\hat{W}\right)$
*denotes the functional introduced in Equation (32) and*
${∥\cdot ∥}_{\mathrm{st}}$
*is the norm introduced in Definition A3.*

**Proof.** In virtue of Lemma 3, we already know the estimate in Equation (40). Making use of Friedrichs–Poincaré inequality ${∥v∥}_{L^{2}\left(\Omega \right)}^{2}\leq C_{P}{∥\nabla v∥}_{L^{2}\left(\Omega \right)}^{2},$ where $C_{P}$ is the domain dependent constant. We see that, if $C_{1}<0$ and $C_{2}<0$, then Equation (40) implies
(43)$$\frac{dV_{\mathrm{neq}}}{dt}\left(\tilde{W}∥\hat{W}\right)\leq -\frac{\left|C_{1}\right|}{C_{P}}{∥\tilde{v}∥}_{L^{2}\left(\Omega \right)}^{2}-\left|C_{2}\right|{∥\tilde{B_{\kappa_{p(t)}}}∥}_{L^{2}\left(\Omega \right)}^{2}.$$
Next, we use Lemma A4, which implies that
(44)$$\left|\tilde{B_{\kappa_{p(t)}}}\right|=\left|\hat{B_{\kappa_{p(t)}}}-B_{\kappa_{p(t)}}\right|\geq {\left|{\hat{B_{\kappa_{p(t)}}}}^{-\frac{1}{2}}\right|}^{-2}\left|I-{\left({\hat{B_{\kappa_{p(t)}}}}^{-\frac{1}{2}}B_{\kappa_{p(t)}}{\hat{B_{\kappa_{p(t)}}}}^{-\frac{1}{2}}\right)}^{\frac{1}{2}}\right|={\left|{\hat{B_{\kappa_{p(t)}}}}^{-\frac{1}{2}}\right|}^{-2}\left|I-Z_{\kappa_{p(t)}}\right|,$$
where $Z_{\kappa_{p(t)}}$ denotes the shifted state introduced in Equation (33). Consequently, we see that
(45)$$-|C_{2}|{∥\tilde{B_{\kappa_{p(t)}}}∥}_{L^{2}\left(\Omega \right)}^{2}=-\left|C_{2}\right|\int_{\Omega }{\left|\tilde{B_{\kappa_{p(t)}}}\right|}^{2}\;dv\leq -\left|C_{2}\right|{\left(\underset{x\in \Omega }{\sup}\left|{\hat{B_{\kappa_{p(t)}}}}^{-\frac{1}{2}}\right|\right)}^{-4}\int_{\Omega }{\left|I-Z_{\kappa_{p(t)}}\right|}^{2}\;dv,$$
which means that Equation (43) can be further manipulated to the form
(46)$$\frac{dV_{\mathrm{neq}}}{dt}\left(\tilde{W}∥\hat{W}\right)\leq -\min\{\frac{\left|C_{1}\right|}{C_{P}},\left|C_{2}\right|{\left(\underset{x\in \Omega }{\sup}|{\hat{B_{\kappa_{p(t)}}}}^{-\frac{1}{2}}|\right)}^{-4}\}({∥\tilde{v}∥}_{L^{2}\left(\Omega \right)}^{2}+{∥I-Z_{\kappa_{p(t)}}∥}_{L^{2}\left(\Omega \right)}^{2}).$$
The inequality in Equation (46) gives in virtue of the definition of the norm ${∥\cdot ∥}_{\mathrm{st}}$ (see Definition A3 and the proposition in Equation (42)). (Recall that the transformation in Equation (33) implies that $\hat{Z_{\kappa_{p(t)}}}=I$.)  □

## 5. Main Result

Using the estimate from Lemma 4 and the relation between the metric and the functional $V_{\mathrm{neq}}$ (see Lemma 1), it is straightforward to prove the following theorem.

**Theorem** **1**(Sufficient conditions for unconditional asymptotic stability)**.**
*Let the pair*
$\hat{W}=\left[\hat{v},\hat{B_{\kappa_{p(t)}}}\right]$
*solve the governing equations for the steady state in Equation (16) with the boundary conditions in Equation (17). If the Reynolds number*
Re*, the Weissenberg number*
Wi
*and the dimensionless shear modulus* Ξ *are such that the constants*
$C_{1}$
*and*
$C_{2}$
*introduced in Equation (41) satisfy*
(47)$$C_{1}<0,\;C_{2}<0,$$
*then the spatially inhomogeneous non-equilibrium steady state*
$\hat{W}$
*is unconditionally asymptotically stable, that is*
(48)$$\mathrm{dist}(\hat{W},W)\overset{t\to +\infty }{\to }0,$$
*holds for* any *initial perturbation*
W*, where the metric*
dist(·,·)
*is the metric introduced in Equation (A7).*

**Proof.** We first investigate the stability in the shifted state space (see Equation (33)), that is we investigate perturbation Z with respect to $\hat{Z}=\left[\hat{v},I\right]$. We introduce the functional $V_{\mathrm{neq}}\left(\tilde{W}∥\hat{W}\right)$ (see Equation (32) and the equivalent expression in Equation (36)). The functional satisfies the condition in Equation (26a) (see Lemma 1). Furthermore, if $C_{1}$ and $C_{2}$ are negative, then Lemma 4 implies that the functional $V_{\mathrm{neq}}\left(\tilde{W}∥\hat{W}\right)$ decreases along the trajectories in a desired manner, that is it satisfies Equation (26b). Consequently, the functional $V_{\mathrm{neq}}\left(\tilde{W}∥\hat{W}\right)$ is a genuine Lyapunov type functional for any neighbourhood of the steady state $\hat{Z}$, hence $\hat{Z}$ is unconditionally asymptotically stable, ${∥\hat{Z}-Z∥}_{\mathrm{st}}\overset{t\to +\infty }{\to }0$.The convergence in the norm on the shifted space seems to be an obscure characterisation of the approach to the equilibrium. However, if we exploit the definition of the shifted state (see Equation (33), and the equality in Equation (A17) proved in Lemma A3), we see that
(49)$${∥\hat{Z}-Z∥}_{\mathrm{st}}={\left({∥\hat{v}-v∥}_{L^{2}\left(\Omega \right)}^{2}+\int_{\Omega }{\left({\mathrm{dist}}_{P(d),\;\mathrm{BW}}(I,{\hat{B_{\kappa_{p(t)}}}}^{-\frac{1}{2}}B_{\kappa_{p(t)}}{\hat{B_{\kappa_{p(t)}}}}^{-\frac{1}{2}})\right)}^{2}\;dv\right)}^{\frac{1}{2}}\geq E\mathrm{dist}(\hat{W},W),$$
where the last inequality follows from the estimate in Equation (A10) in Lemma A2. Here, *E* is a positive constant that depends on $\hat{W}$, and dist(·,·) is the metric introduced in Definition A2 (Equation (A7)). *This metric is a natural one if we restrict ourselves to the set of positive definite tensor fields*. The inequality in Equation (49) then implies Equation (48).  □

We note that if we want to investigate the spatially homogeneous steady state $\hat{W}=\left[\hat{v},\hat{B_{\kappa_{p(t)}}}\right]{=}_{\mathrm{def}}\left[0,I\right]$, that is if we set the boundary condition V=0, then
(50)C1=−1Re,C2=−αΞ2Wi,
and the condition in Equation (47) is *automatically satisfied without any restriction of the values of Reynolds number and the Weissenberg number*. If the steady state is non-trivial, that is if $\hat{W}=\left[\hat{v},\hat{B_{\kappa_{p(t)}}}\right]\neq \left[0,I\right]$, then the condition in Equation (47) must be evaluated. This is done in Section 6 for the Taylor–Couette flow. (Note that $\hat{B_{\kappa_{p(t)}}}$ and $\hat{v}$ are solutions to Equation (16), hence they depend on the Weissenberg and Reynolds number.) Naturally, one can expect that the condition will hold for a sufficiently small Reynolds number and Weissenberg number.

Having the Lyapunov type functional given by Equation (32), it is interesting to see how the functional works in the case of close to the equilibrium setting, that is for $\hat{B_{\kappa_{p(t)}}}\approx I$, and for small perturbations, that is for small $\tilde{B_{\kappa_{p(t)}}}$. We see that if $\tilde{B_{\kappa_{p(t)}}}$ is small and if $\hat{B_{\kappa_{p(t)}}}$ is close to the identity, then
(51)$$-\ln\det(I+{\hat{B_{\kappa_{p(t)}}}}^{-1}\tilde{B_{\kappa_{p(t)}}})+\mathrm{Tr}({\hat{B_{\kappa_{p(t)}}}}^{-1}\tilde{B_{\kappa_{p(t)}}})\approx \frac{1}{2}\mathrm{Tr}({\left({\hat{B_{\kappa_{p(t)}}}}^{-1}\tilde{B_{\kappa_{p(t)}}}\right)}^{2})\approx \frac{1}{2}{\left|\tilde{B_{\kappa_{p(t)}}}\right|}^{2},$$
and the proposed Lyapunov type functional $V_{\mathrm{neq}}$ can be approximated as $V_{\mathrm{neq}}\approx V_{\mathrm{naive}}$ where
(52)$$V_{\mathrm{naive}}(\tilde{W}∥\hat{W}){=}_{\mathrm{def}}\int_{\Omega }\frac{1}{2}{\left|\tilde{v}\right|}^{2}\;dv+\int_{\Omega }\frac{\Xi }{4}{\left|\tilde{B_{\kappa_{p(t)}}}\right|}^{2}\;dv.$$

The functional $V_{\mathrm{naive}}$ might be the first candidate for the Lyapunov type functional if the stability is investigated using the popular “energy method” (see, for example, Straughan [57]). The functional is clearly nonnegative, and it vanishes if and only if the perturbation vanishes. Moreover, the candidate $V_{\mathrm{naive}}$ for the Lyapunov type functional is much simpler than $V_{\mathrm{neq}}$. Indeed, the proximity of the perturbation to the non-equilibrium state $\left[\hat{v},\hat{B_{\kappa_{p(t)}}}\right]$ is measured using the standard Lebesgue space norms, and $V_{\mathrm{naive}}$ does not depend on the value of $\hat{B_{\kappa_{p(t)}}}$. Therefore, it seems that $V_{\mathrm{naive}}$ is a good candidate for the Lyapunov type functional for the analysis of arbitrary spatially inhomogeneous non-equilibrium steady state $\left[\hat{v},\hat{B_{\kappa_{p(t)}}}\right]$.

The inappropriateness of $V_{\mathrm{naive}}$ for the stability analysis is in fact apparent even in a very trivial setting. Let us consider the *spatially homogeneous equilibrium steady state*
$\hat{B_{\kappa_{p(t)}}}=I$, $\hat{v}=0$ in a mechanically isolated container, that is we set V=0 in the boundary condition in Equation (12). If we use the (exact) evolution equations for the perturbation velocity in Equation (A28b), and if we evaluate the time derivative of $V_{\mathrm{naive}}$, then we get
dVnaivedt(W˜∥W^)=∫Ωv˜•∂v˜∂t dv+∫ΩΞ2Tr(Bκp(t)˜∂Bκp(t)˜∂t) dv(53)=−∫Ω2ReD˜:D˜ dv−∫ΩΞ Bκp(t)˜:D˜ dv−∫ΩD^v˜•v˜ dv+∫ΩΞ2Tr(Bκp(t)˜∂Bκp(t)˜∂t) dv,
(see also Equation (A34)). The last term on the right-hand side of Equation (53) can be evaluated using the (exact) evolution equation for $\tilde{B_{\kappa_{p(t)}}}$ (see Equation (A29)). Substituting Equation (A29) into Equation (53) and using the fact that $\hat{B_{\kappa_{p(t)}}}=I$ and $\hat{v}=0$, yields
dVnaivedt(W˜∥W^)=−∫Ω2ReD˜:D˜ dv−∫ΩΞ2Tr[({v˜•∇}Bκp(t)˜)Bκp(t)˜] dv+∫ΩΞTr(D˜Bκp(t)˜2) dv(54)−∫ΩΞ2WiTr[αBκp(t)˜3+2αBκp(t)˜2+(1−2α)Bκp(t)˜2] dv.
Using the standard manipulation
(55)$$\int_{\Omega }\frac{\Xi }{2}\mathrm{Tr}[(\{\tilde{v}•\nabla \}\tilde{B_{\kappa_{p(t)}}})\tilde{B_{\kappa_{p(t)}}}]\;dv=\int_{\partial \Omega }\frac{\Xi }{4}{\left|\tilde{B_{\kappa_{p(t)}}}\right|}^{2}(\tilde{v}•n)\;ds-\int_{\Omega }\frac{\Xi }{4}(\mathrm{div}\tilde{v}){\left|\tilde{B_{\kappa_{p(t)}}}\right|}^{2}\;dv,$$
we see that the second term on the right-hand side of Equation (54) vanishes in virtue of the incompressibility constraint in Equation (A28a) and the boundary condition for $\tilde{v}$. Consequently, Equation (54) reduces to
dVnaivedt(W˜∥W^)=−∫Ω2ReD˜:D˜ dv+∫ΩΞTr(D˜Bκp(t)˜2) dv(56)−∫ΩΞ2WiTr[αBκp(t)˜3+2αBκp(t)˜2+(1−2α)Bκp(t)˜2] dv.

Let us now consider an initial perturbation that is chosen is such a way that $\int_{\Omega }\Xi \mathrm{Tr}\left(\tilde{D}{\tilde{B_{\kappa_{p(t)}}}}^{2}\right)\;dv>0$, which can certainly be done. This positive value will dominate the right-hand side of Equation (56) provided that the Reynolds number and the Weissenberg number are large enough. Consequently, $V_{\mathrm{naive}}$ will (initially) increase, and it would be useless as the Lyapunov type functional unless we a priori limit ourselves to small perturbations.

On the other hand, if we use the functional $V_{\mathrm{neq}}$ in the case $\hat{B_{\kappa_{p(t)}}}=I$ and $\hat{v}=0$, then we *immediately* see that the constants $C_{1}$ and $C_{2}$ in Equation (40) are negative, and that the equilibrium steady state is asymptotically stable with respect to *any* perturbation and *any* value of the Reynolds and the Weissenberg number! (Note that Guillopé and Saut [58] obtained only a conditional stability result in a Sobolev space norm for the equilibrium rest state $\hat{B_{\kappa_{p(t)}}}=I$ and $\hat{v}=0$ (see their Corollary 3.5 and assumptions of Theorem 3.3).) Based on the analysis presented above, we can therefore claim that we have indeed benefited from a *well constructed Lyapunov type functional*
$V_{\mathrm{neq}}$
*and the choice of metric*. Unlike the naive Lyapunov type functional $V_{\mathrm{naive}}$, the proposed Lyapunov type functional $V_{\mathrm{neq}}$ seems to properly reflect the nonlinearity of the governing equations and the related energy storage mechanisms and the entropy production mechanisms.

## 6. Taylor–Couette Flow

Let us now consider a viscoelastic fluid described by the Giesekus model introduced in Section 2 with $\alpha =\frac{1}{2}$, and let us investigate the stability of steady flow in the standard Taylor–Couette flow geometry (see Figure 1). The objective is to show as how the theory introduced above works in a specific setting. The choice $\alpha =\frac{1}{2}$ is motivated by the simplicity of the expressions for the corresponding steady state.

The fluid is placed in between two infinite concentric cylinders of radii $R_{1}$ and $R_{2}$, with $R_{1}<R_{2}$. The cylinders are rotating with the angular velocities $\Omega_{1}$ (inner cylinder) and $\Omega_{2}$ (outer cylinder) along the common axis. The geometry naturally leads to the use of cylindrical coordinates (r,φ,z); the normed basis vectors are denoted as $g_{\hat{r}}$, $g_{\hat{\varphi }}$ and $g_{\hat{z}}$ (see Figure 1). Since the domain is unbounded in the *z*-direction, we henceforth consider a periodic cell
(57)$$\Omega {=}_{\mathrm{def}}\left\{\left(r,\varphi ,z\right)\in R^{3}\;|\;R_{1}<r<R_{2},\;0\leq \varphi <2\pi ,\;\left|z\right|<h\right\}$$
where h>0 is arbitrary, and we use the notation $\Gamma_{1}{=}_{\mathrm{def}}\left\{\left(r,\varphi ,z\right)\in R^{3}\;|\;R_{1}<r<R_{2},\;0\leq \varphi <2\pi ,\;\left|z\right|=h\right\}$ for the top and bottom base, and $\Gamma_{2}{=}_{\mathrm{def}}\left\{\left(r,\varphi ,z\right)\in R^{3}\;|\;r\in \left\{R_{1},R_{2}\right\},\;0\leq \varphi <2\pi ,\;\left|z\right|<h\right\}$ for the cylindrical walls of the domain. The flow is driven by the rotation of the cylinders along the common axis.

### 6.1. Base Flow—Non-Equilibrium Steady State

The first task in the stability analysis is to find the steady solution to the governing equations. This solution is the spatially inhomogeneous non-equilibrium steady state $\hat{W}$ as introduced in Section 3.2. The characteristic length and characteristic time have been chosen as $x_{\mathrm{char}}{=}_{\mathrm{def}}R_{1}$, $t_{\mathrm{char}}{=}_{\mathrm{def}}\frac{1}{\Omega_{1}}$. We use the periodic boundary condition on $\Gamma_{1}$ and the no-slip boundary condition for velocity field v on $\Gamma_{2}$, that is ${v|}_{r=R_{1}}=R_{1}\Omega_{1}g_{\hat{\varphi }}$, ${v|}_{r=R_{2}}=R_{2}\Omega_{2}g_{\hat{\varphi }}$. These boundary conditions are consistent with the requirements on boundary conditions specified in Section 2.4. In their dimensionless form, the boundary conditions read
(58)$${v^{⋆}|}_{r^{⋆}=1}=g_{\hat{\varphi }},\;{v^{⋆}|}_{r^{⋆}=\frac{1}{\eta }}=\frac{\zeta }{\eta }g_{\hat{\varphi }},$$
where we have introduced two dimensionless parameters $\eta {=}_{\mathrm{def}}\frac{R_{1}}{R_{2}}$ and $\zeta {=}_{\mathrm{def}}\frac{\Omega_{2}}{\Omega_{1}}$. Hereafter, we work exclusively with the dimensionless variables and thus, for the sake of simplicity, we omit the star denoting them.

Since the problem has the rotational symmetry, we search for the steady non-equilibrium state in a special form. Namely, the solution to Equation (16) subject to boundary conditions in Equation (58) is sought in the form
(59)$$\hat{v}=v^{\hat{\varphi }}\left(r\right)g_{\hat{\varphi }},\;\hat{m}=\hat{m}\left(r\right),\;\hat{B_{\kappa_{p(t)}}}=[\begin{array}{ccc} {B^{\hat{r}}}_{\hat{r}}\left(r\right) & {B^{\hat{r}}}_{\hat{\varphi }}\left(r\right) & 0\\ {B^{\hat{\varphi }}}_{\hat{r}}\left(r\right) & {B^{\hat{\varphi }}}_{\hat{\varphi }}\left(r\right) & 0\\ 0 & 0 & {B^{\hat{z}}}_{\hat{z}}\left(r\right) \end{array}]$$
Note that the chosen ansatz for the velocity field automatically satisfies the incompressibility condition. The assumptions lead to the following expressions for the velocity gradient, the symmetric part of the velocity gradient, the convective term, the divergence of $\hat{B_{\kappa_{p(t)}}}$, and the upper convected derivative of $\hat{B_{\kappa_{p(t)}}}$:
∇v^=[0−ω0rdωdr+ω00000],D^=[0r2dωdr0r2dωdr00000],dv^dt=[−rω200],(60)divBκp(t)^=[1rddr(rBr^r^)−Bφ^φ^rdBφ^r^dr+Bφ^r^+Br^φ^r0],Bκp(t)^¯▿=[0−rdωdrBr^r^0−rdωdrBr^r^−2rdωdrBφ^r^0000],
where we introduce the angular velocity ω(r), $v^{\hat{\varphi }}\left(r\right){=}_{\mathrm{def}}\omega \left(r\right)r$. Using Equation (60), we see that the governing equations for the velocity field in Equation (16b) reduce to
(61a)[−rω200]=[ddr(m^+Ξ(Br^r^−13(Br^r^+Bφ^φ^+Bz^z^)))+ΞBr^r^−Bφ^φ^r1r2ddr(1Rer3dωdr+Ξr2BBr^φ^)0],
while the governing equations in Equation (16c) for $\hat{B_{\kappa_{p(t)}}}$ read
[0−rdωdrBr^r^0−rdωdrBr^r^−2rdωdrBφ^r^0000]=(61b) −1Wi[α((Br^r^)2+(Br^φ^)2)+(1−2α)Br^r^−(1−α)αBr^φ^(Br^r^+Bφ^φ^)+(1−2α)Br^φ^0αBr^φ^(Br^r^+Bφ^φ^)+(1−2α)Br^φ^α((Br^φ^)2+(Bφ^φ^)2)+(1−2α)Bφ^φ^−(1−α)000α(Bz^z^)2+(1−2α)Bz^z^−(1−α)].
Assuming that $\frac{d\omega }{dr}\neq 0$ in $\left(R_{1},R_{2}\right)$, Equation (61b) can be solved for ${B^{\hat{r}}}_{\hat{r}}$, ${B^{\hat{r}}}_{\hat{\varphi }}$, ${B^{\hat{\varphi }}}_{\hat{\varphi }}$ and ${B^{\hat{z}}}_{\hat{z}}$. However, for general α∈(0,1), the formulae for the aforementioned quantities are too complex to be written down here. Let us note however that for $\alpha =\frac{1}{2}$ the formulae simplify significantly; the solution to Equation (61b) which satisfies the condition of $\hat{B_{\kappa_{p(t)}}}$ being positive definite in this case reads
(62)$${B^{\hat{z}}}_{\hat{z}}=1,\;{B^{\hat{r}}}_{\hat{\varphi }}=\frac{-1+\sqrt{1+c^{2}}}{c},\;{B^{\hat{r}}}_{\hat{r}}=\sqrt{\frac{2}{c}{B^{\hat{r}}}_{\hat{\varphi }}},\;{B^{\hat{\varphi }}}_{\hat{\varphi }}=\sqrt{2(c+\frac{1}{c}){B^{\hat{r}}}_{\hat{\varphi }}},$$
where we denote $c{=}_{\mathrm{def}}2\mathrm{Wi}\;r\frac{d\omega }{dr}$. Substituting Equation (62) into the second equation in Equation (61a) then yields an ordinary differential equation for the angular velocity ω
(63)0=ddr(1Rer3dωdr+Ξr2−1+1+4Wi2 r2(dωdr)22Wi rdωdr),
supplemented by the boundary conditions ${\omega |}_{r=1}=1$, ${\omega |}_{r=\frac{1}{\eta }}=\zeta$, which follow from Equation (58) and the fact that $v^{\hat{\varphi }}\left(r\right)=\omega \left(r\right)r$. Equation (63) together with the boundary conditions constitute a boundary value problem which needs to be solved numerically.

### 6.2. Explicit Criterion for the Stability of Spatially Inhomogeneous Non-Equilibrium Steady State

Here, we explicitly compute constants $C_{1}$, $C_{2}$ defined by Equations (41a) and (41b) for the Taylor–Couette problem and for the specific values of the dimensionless numbers Ξ, Re and Wi. Let us recall that, for the sake of simplicity, we set $\alpha =\frac{1}{2}$, and we consider the steady tensor field $\hat{B_{\kappa_{p(t)}}}$ given by Equation (62). We fix the values for the geometric parameter η and angular velocities ratio ζ as $\eta =\frac{1}{2}$ and ζ=2.

The angular velocity ω is obtained by solving Equation (63) which is a boundary-value problem for a second order nonlinear differential equation. The problem was solved numerically using the function solve_bvp from SciPy library version 1.0.0, which implements a fourth-order collocation algorithm with the control of residuals as described in Kierzenka and Shampine [59]. With the angular velocity ω in hand, we immediately get the steady velocity field $\hat{v}=\omega \left(r\right)rg_{\hat{\varphi }}$, and the steady left Cauchy–Green tensor field $\hat{B_{\kappa_{p(t)}}}$ through Equation (62). The plots of the velocity field and the components of $\hat{B_{\kappa_{p(t)}}}$ are shown in Figure 2.

Having computed the steady velocity field $\hat{v}$ and the corresponding steady field $\hat{B_{\kappa_{p(t)}}}$, we can evaluate the constants $C_{1}$ and $C_{2}$ in the estimate in Equation (40). The gradient of $\hat{v}$ as well as the gradient of $\hat{B_{\kappa_{p(t)}}}$ are again computed numerically from the obtained numerical solution. Finally, the Poincaré constant for the cylindrical annulus is determined via an explicit solution of the corresponding eigenvalue problem −Δu=λu for the Laplace operator with Dirichlet boundary condition, which leads, for the geometrical parameter $\eta =\frac{1}{2}$, to the value $C_{P}\approx 0.1025.$ The resulting stability regions in the Re–Wi plane are shown in Figure 3 for a fixed value of the dimensionless shear modulus Ξ. As one might expect the spatially inhomogeneous steady state is indeed unconditionally asymptotically stable if the Weissenberg number Wi and the Reynolds number Re are small enough.

### 6.3. Numerical Experiments—Evolution of Various Initial Perturbations

Finally, we document the theoretically predicted behaviour by numerical experiments. The numerical experiments allow us to quantitatively track the evolution of key quantities such as the net kinetic energy, and also to quantitatively monitor the energy exchange between the fluid and its surroundings.

The governing equations were numerically solved using standard techniques. The weak forms of the governing equations were discretised in the space using the finite element method, while the time derivatives were approximated with the backward Euler method. The two-dimensional domain Ω was discretised by regular quadrilaterals. The mesh divided the annular region Ω into 80 pieces in the radial direction, and in 720 pieces in the azimuthal direction. The corresponding total number of degrees of freedom in all numerical experiments was over $1.3\times 10^{6}$. The velocity field v and the $B_{\kappa_{p(t)}}$ field were approximated by biquadratic Q2 elements, and the pressure field *m* was approximated by the piecewise linear discontinuous elements ${P1}^{d}$ (see Korelc and Wriggers [60] for details). The finite element pair that was used for the velocity/pressure fields satisfied the Babuška-Brezzi condition, the finite element for $B_{\kappa_{p(t)}}$ field was chosen to be the same as for the velocity in order to provide rich enough finite element space for the solution. The same finite elements were chosen for the two-dimensional simulation of other viscoelastic rate-type fluids (Oldroyd, Burgers and their various nonlinear versions) by Hron et al. [61] and Málek et al. [62]. In the three-dimensional case, low order elements can be used to decrease the overall cost of the calculation (see Tůma et al. [63]).

The numerical scheme was implemented in the AceGen/AceFEM system (see Korelc [64] and Korelc [65]). The main advantage of the system is that it provides automatic differentiation used for the computation of the exact tangent matrix needed by the Newton solver that treats all nonlinearities. The final set of linear equations was solved by the direct solver Intel MKL Pardiso. The stopping criterion for the Newton solver was set to $10^{-9}$.

Using the numerical scheme, we are ready to study the behaviour of various perturbations to the non-equilibrium steady state. In all scenarios described below, we use the dimensionless parameters
(64)Ξ=0.1,Re=50,Wi=5,α=12
and we fix the geometric parameter η and angular velocities ratio ζ as in Section 6.2, that is $\eta =\frac{1}{2}$ and ζ=2. The chosen values of η, ζ and Ξ correspond to the stability diagram shown in Figure 3a. The values of Reynolds number and Weissenberg number are outside the region where we have *proven* the decay of the proposed Lyapunov type functional. Nevertheless, as we show below, the Lyapunov type functional is, in the cases being investigated below, still a decreasing function.

First, we start from the homogeneous steady state solution $\left[v,B_{\kappa_{p(t)}},m\right]=\left[0,I,0\right]$, and we let the system to spontaneously evolve up to the time instant t=1000. (More precisely, the initial condition is v=0 inside the domain Ω, and Equation (17) holds on the boundary of Ω. After the first computational time step, which is chosen as Δt=0.05, we get on the discrete level a divergence-free velocity field with the appropriate boundary condition. This discrete velocity field provides us a consistent initial condition for further computations. Therefore, we formally start the evolution not at t=0, but at t=0.05.) At this time instant, the system is almost relaxed and is close to the steady solution. The solution at t=1000 is used as a starting point for solving the steady governing equations (without the time derivatives) and the spatially inhomogeneous non-equilibrium *steady* state is obtained just in two Newton iterations. (The finite element solution coincides with the semi-analytical steady solution obtained in Section 6.1. This among others provides us a tool for the code verification.) Consequently, the finite element solution is in what follows used as the spatially inhomogeneous non-equilibrium steady state $\hat{W}$.

Having obtained the numerical representation of the spatially inhomogeneous non-equilibrium steady state, we proceed with two scenarios concerning the specification of the initial perturbation.

#### 6.3.1. Scenario A—Localised Perturbation of the Left Cauchy–Green Field

In the first scenario, we keep the initial velocity field perturbation equal to zero,
(65)$${\tilde{v}|}_{t=0}=0,$$
while the initial perturbation in $B_{\kappa_{p(t)}}$ is localised in space (see the first snapshot in Figure 4). Since the system is fully coupled, the perturbation in the $B_{\kappa_{p(t)}}$ field triggers for t>0 a nontrivial evolution of the velocity perturbation $\tilde{v}$ (see Figure 5). This can be observed also in the plots showing the evolution of the net elastic stored energy and the net kinetic energy (see Figure 6).

Finally, we also investigate the time evolution of the proposed Lyapunov type functional $V_{\mathrm{neq}}$ and the naive Lyapunov type functional $V_{\mathrm{naive}}$, and the net mechanical energy flux going through the boundary of Ω (see Figure 6). Although we work with the Reynolds number/Weissenberg number pair outside the guaranteed stability region, we see that the value of Lyapunov type functional $V_{\mathrm{neq}}$ still decreases in time, and that the perturbation vanishes for t→+∞. This indicates that the estimates on the time derivative of the proposed Lyapunov type functional are, at least for a class of perturbations, too strict and they might be improved. One should also note that the “net kinetic energy” of the perturbation, that is the functional $\int_{\Omega }\frac{1}{2}{\left|\tilde{v}\right|}^{2}\;dv$, *does not* decrease for all t>0 (see Figure 6b). In fact, it experiences a transitional growth, and such a transient growth can be observed even for the Reynolds number/Weissenberg number values within the stability region. This is a natural observation. The elastic energy stored in the material can be released in the form of the kinetic energy. It is only the combination of the elastic energy and the kinetic energy that appears in the Lyapunov type functional that leads to a quantity that decays at any time.

Further, the net mechanical energy flux through the boundary fluctuates around the value that corresponds to the non-equilibrium steady state, and then it reaches the value that corresponds to the spatially inhomogeneous non-equilibrium steady state (see Figure 6d). This can again happen even if the Reynolds number/Weissenberg number take values within the stability region.

#### 6.3.2. Scenario B—Global Perturbation of the Velocity Field

In the second scenario, we start with a nonzero velocity perturbation $\tilde{v}$, and the $B_{\kappa_{p(t)}}$ field is kept unchanged,
(66)$${\tilde{B_{\kappa_{p(t)}}}|}_{t=0}=O.$$
The initial velocity v is prescribed as
(67)$${v|}_{t=0}=\Omega rg_{\hat{\varphi }},$$
where the angular velocity is the arithmetic mean of the two angular velocities $\Omega =\frac{\Omega_{1}+\Omega_{2}}{2}$. (Formally, we apply the same procedure as discussed in the previous section. The initial condition is $v=\Omega rg_{\hat{\varphi }}$ inside the domain Ω, and Equation (17) holds on the boundary of Ω. The actual computation starts after the first (formal) time step, when the discrete velocity field is divergence-free and it fulfills the boundary condition.) Again, as in the previous case, the non-zero perturbation in one unknown field ($\tilde{v}$) triggers for t>0 a nontrivial evolution of the other unknown field ($\tilde{B_{\kappa_{p(t)}}}$) (see Figure 7 and Figure 8).

Moreover, this numerical experiment is instructive for yet another reason. In Figure 9c, we plot the time evolution of the values of the functionals $V_{\mathrm{neq}}$ and $V_{\mathrm{naive}}$. Clearly, the functional $V_{\mathrm{naive}}$ (see Equation (52)), which is a naive candidate for the Lyapunov type functional, experiences a transitional growth. Interestingly, the proposed complex Lyapunov type functional $V_{\mathrm{neq}}$ is still a decreasing function, although the Reynolds number/Weissenberg number values are outside the region, where we have actually proven the decay of the functional. This further indicates that the functional $V_{\mathrm{naive}}$ is indeed not a good candidate for a Lyapunov type functional (see also Section 5 for further discussion).

## 7. Conclusions

We have investigated the stability of spatially inhomogeneous non-equilibrium steady states (flows) of viscoelastic fluids described by the Giesekus model. We have derived bounds on the values of the Reynolds number and the Weissenberg number that guarantee the flow stability subject to *any finite perturbation*. The stability has been investigated using the Lyapunov type functional given by the formula
(68)$$V_{\mathrm{neq}}(\tilde{W}∥\hat{W})=\int_{\Omega }\frac{1}{2}{\left|\tilde{v}\right|}^{2}\;dv+\int_{\Omega }\frac{\Xi }{2}[-\ln\det(I+{\hat{B_{\kappa_{p(t)}}}}^{-1}\tilde{B_{\kappa_{p(t)}}})+\mathrm{Tr}({\hat{B_{\kappa_{p(t)}}}}^{-1}\tilde{B_{\kappa_{p(t)}}})]\;dv.$$

A few observations concerning the proposed Lyapunov type functional are at hand.

First, the proposed Lyapunov type functional has a relatively complicated form. In particular, it is *not* quadratic in the perturbation $\tilde{B_{\kappa_{p(t)}}}$, and it *depends* on the spatially inhomogeneous non-equilibrium state $\hat{B_{\kappa_{p(t)}}}$. This makes it remarkably different from a naive Lyapunov type functional of the form
(69)$$V_{\mathrm{naive}}(\tilde{W}∥\hat{W})=\int_{\Omega }\frac{1}{2}{\left|\tilde{v}\right|}^{2}\;dv+\int_{\Omega }\frac{\Xi }{4}{\left|\tilde{B_{\kappa_{p(t)}}}\right|}^{2}\;dv,$$
which might be a first try if the stability problem were analysed using the popular “energy method”. However, as we have shown, the complicated structure of the proposed functional $V_{\mathrm{neq}}$ leads to a relatively simple and well structured formula for its time derivative, which in turn allows one to formulate conditions that guarantee the negativity of the time derivative. Furthermore, the complicated structure of the proposed functional $V_{\mathrm{neq}}$ also leads to a simple relation between the functional and the metric
(70)$$\mathrm{dist}(\hat{W},W){=}_{\mathrm{def}}{\left({∥\hat{v}-v∥}_{L^{2}\left(\Omega \right)}^{2}+{\left[{\mathrm{dist}}_{P_{\Omega }\left(d\right),\;\mathrm{BW}}(\hat{B_{\kappa_{p(t)}}},B_{\kappa_{p(t)}})\right]}^{2}\right)}^{\frac{1}{2}}$$
on the set of spatially distributed symmetric positive definite matrices.

Second, the Lyapunov type functional has been used in the investigation of stability of solution to the *complete system of nonlinear governing equations*. In particular, the evolution equations for the perturbation have been investigated without any simplification. This makes the present approach different from the “energy budget” analysis (see, for example, Joo and Shaqfeh [66], Byars et al. [67], Ganpule and Khomami [68], Smith et al. [69], Karapetsas and Tsamopoulos [70], Pettas et al. [71], and especially the work by Grillet et al. [19] who investigated the Giesekus model). The “energy budget” analysis, although valuable in the discussion of the nature of the instability mechanisms, is based on the *linearised* momentum equation for the perturbation and *linearised* constitutive equation for the “polymeric stress”. Consequently, the standard “energy budget” analysis is, unlike the present approach, a tool that cannot be used in the finite amplitude stability analysis of the *complete system of nonlinear governing equations*. One might also note that, despite the complexity of the proposed Lyapunov type functional, the formula for its time derivative is in fact quite simple compared to the formulae in the “energy budget” analysis. This happens even though the “energy budget” formulae paradoxically stem from various simplifications of the original system of governing equations.

Third, the Lyapunov type functional has been designed using thermodynamical arguments. In fact, the proposed Lyapunov type functional has been constructed from the net mechanical energy functional $E_{\mathrm{mech}}$ (see Equation (10), via Equation (30)). This makes the construction quite general, and one might speculate that a similar approach is very likely applicable to other popular viscoelastic rate-type models such as the PTT model (see Phan Thien and Tanner [72]) or the FENE-P model (see Bird et al. [73]), as well as complex viscoelastic rate-type models with, for example, stress diffusion terms (see Málek et al. [37] or Dostalík et al. [39]). Further, the construction of the Lyapunov type functional has been based on the method proposed by Bulíček et al. [23], and this method is speculated to work even for complex coupled thermo-mechanical systems. This naturally calls for the investigation of the applicability of the method in more complex settings such as flows of viscoelastic rate-type fluids with temperature dependent material parameters.

Fourth, thermodynamical type considerations such as the identification of the energy storage mechanisms and entropy producing mechanisms are known to play an important role in the rigorous mathematical theory of nonlinear partial differential equations governing the motion of viscoelastic fluids (see, for example, Hu and Lelièvre [74], Boyaval et al. [40], Barrett and Boyaval [75], Barrett and Boyaval [76], Barrett and Süli [47] or Bulíček et al. [77]). On the other hand, rigorous mathematical analysis of *long-time behaviour* of viscoelastic fluids is usually done without a direct appeal to thermodynamics, and the available results are quite limited especially if one considers thermodynamically open systems (see, for example, Guillopé and Saut [78], Nohel and Pego [79], Jourdain et al. [80] or Renardy [81]). (Usually, only stability of unidirectional steady flows in simple geometries is considered.) Consequently, the approach proposed in the current contribution might be of interest from the rigorous mathematical perspective as well. This means that one should deal with the weak solution to the governing equations, and that one should investigate the applicability of the presented arguments for a solution/perturbation that has only the smoothness that can be actually proven.

## Figures and Tables

**Figure 1 entropy-21-01219-f001:**
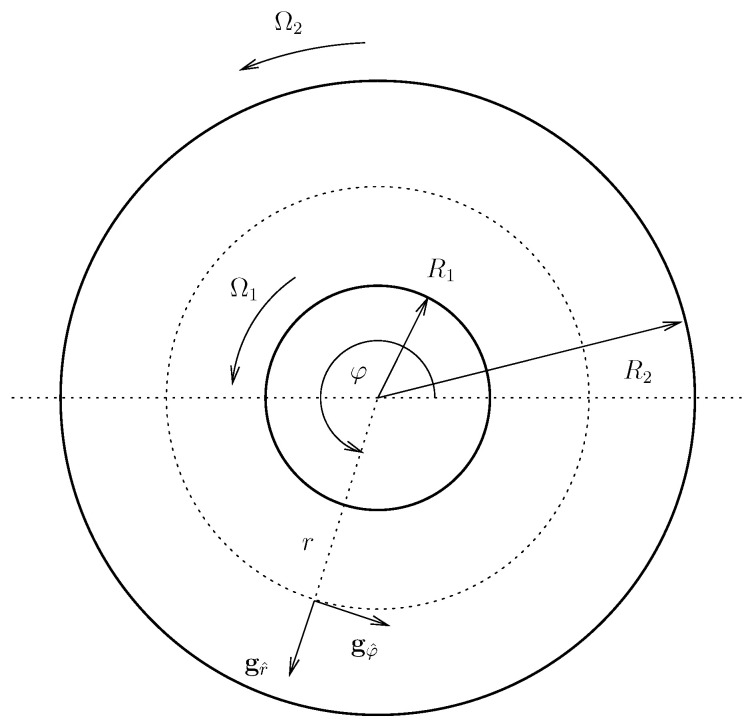
Cylindrical Taylor–Couette flow.

**Figure 2 entropy-21-01219-f002:**
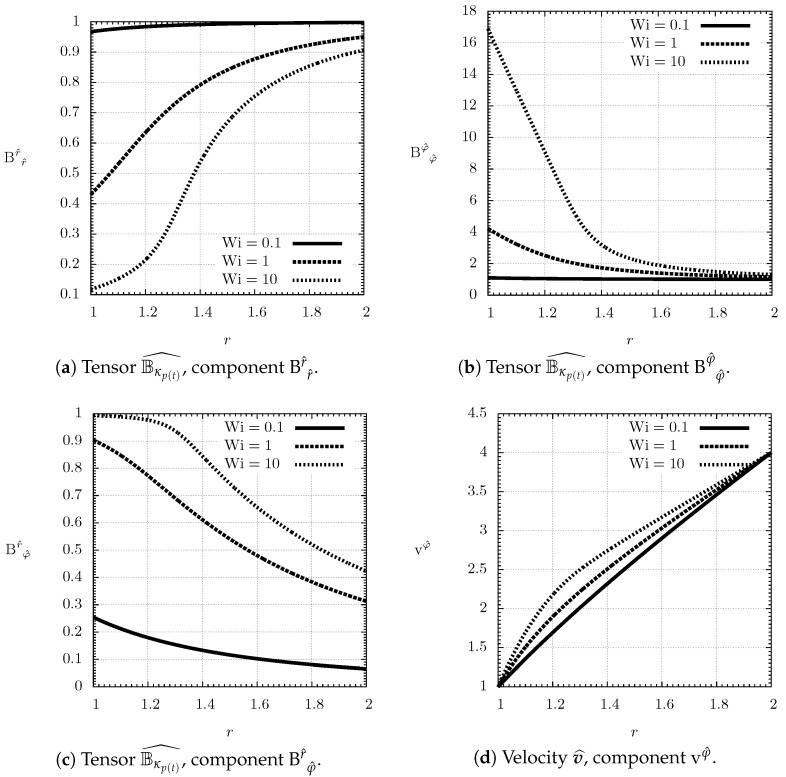
Taylor–Couette flow, spatially inhomogeneous non-equilibrium steady state for various values of Weissenberg number Wi, Giesekus parameter $\alpha =\frac{1}{2}$, Reynolds number Re=100, dimensionless shear modulus Ξ=0.1 and problem parameters $\eta =\frac{1}{2}$ and ζ=2.

**Figure 3 entropy-21-01219-f003:**
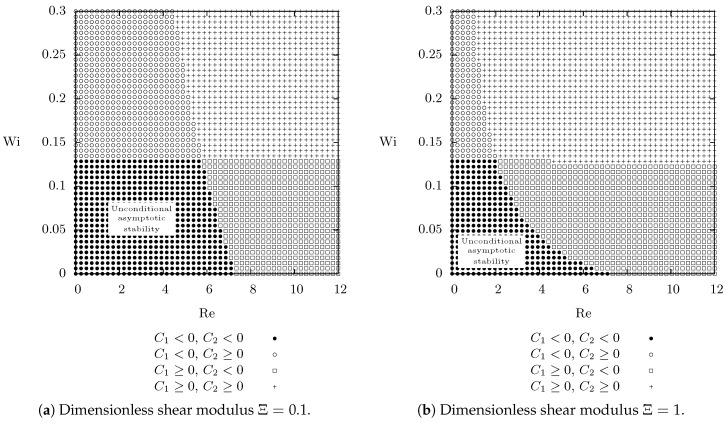
Taylor–Couette flow, numerical values of constants $C_{1}$ and $C_{2}$ for various values of the Reynolds number Re, Weissenberg number Wi and the dimensionless shear modulus Ξ. *Unconditional asymptotic stability is granted provided that*
$C_{1}<0$
*and*
$C_{2}<0$, numerical values of constants $C_{1}$ and $C_{2}$ are evaluated using Equation (41). Giesekus parameter $\alpha =\frac{1}{2}$ and problem parameters $\eta =\frac{1}{2}$ and ζ=2.

**Figure 4 entropy-21-01219-f004:**
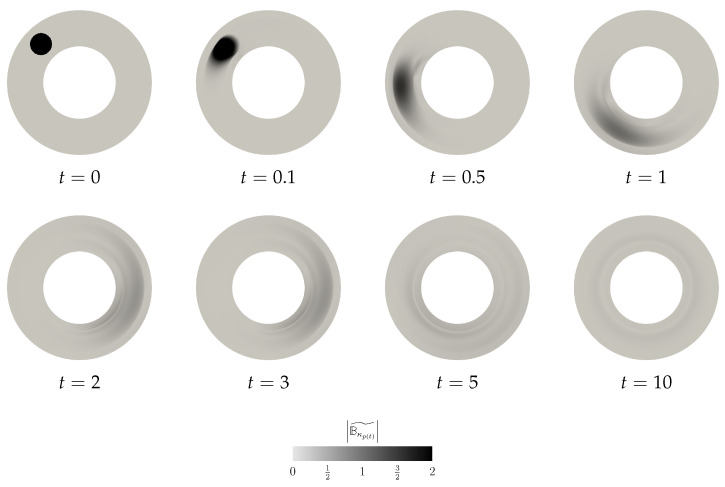
Scenario A, snapshots of $\left|\tilde{B_{\kappa_{p(t)}}}\right|$ at different time instants.

**Figure 5 entropy-21-01219-f005:**
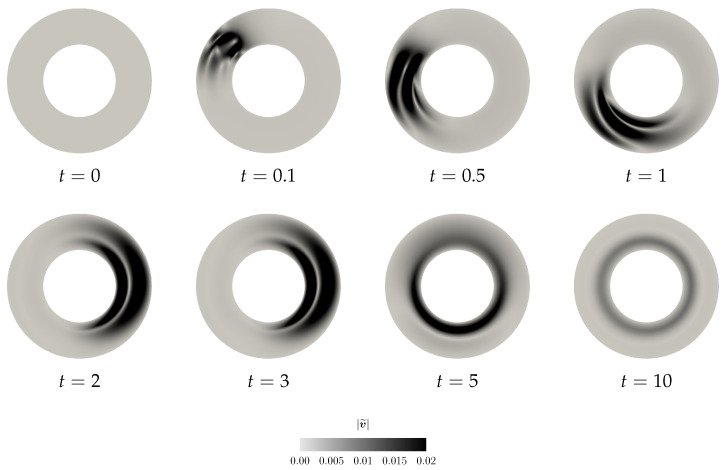
Scenario A, snapshots of $\left|\tilde{v}\right|$ at different time instants.

**Figure 6 entropy-21-01219-f006:**
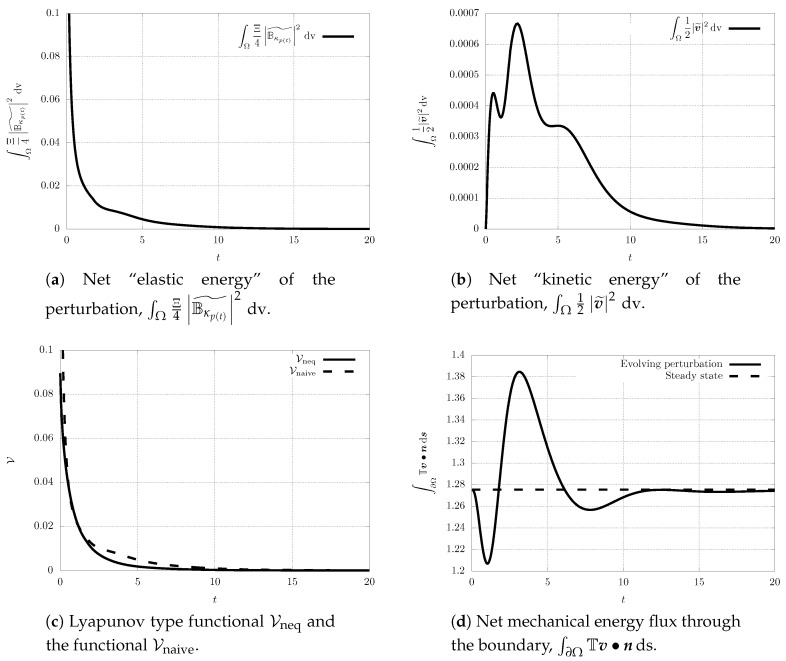
Scenario A, time evolution of the net quantities.

**Figure 7 entropy-21-01219-f007:**
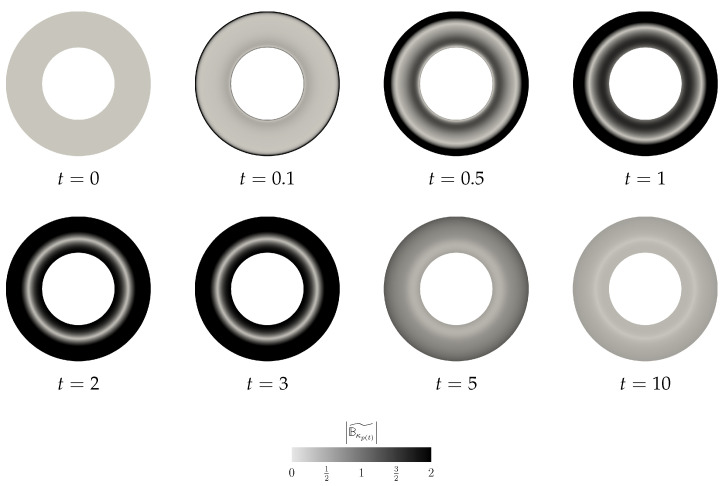
Scenario B, snapshots of $\left|\tilde{B_{\kappa_{p(t)}}}\right|$ at different time instants.

**Figure 8 entropy-21-01219-f008:**
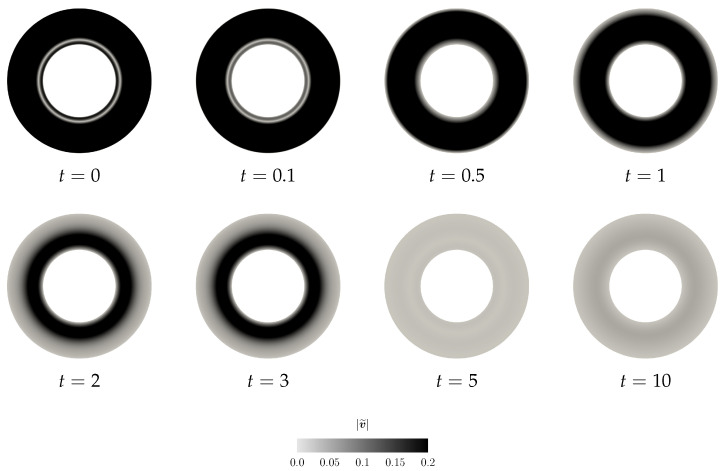
Scenario B, snapshots of $\left|\tilde{v}\right|$ at different time instants.

**Figure 9 entropy-21-01219-f009:**
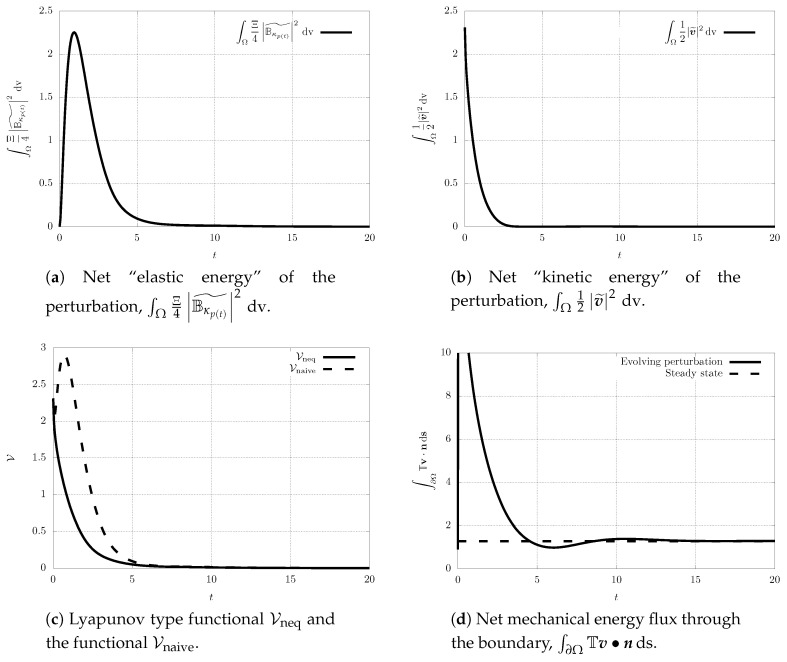
Scenario B, time evolution of the net quantities.

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
