# Peer review of "Finite Amplitude Stability of Internal Steady Flows of the Giesekus Viscoelastic Rate-Type Fluid"

_entropy, 2019, doi:10.3390/e21121219_

Round 1

Reviewer 1 Report

The Authors presentation of the problem, used tools in examining the problem and obtained results is clear and very well argued. Theoretical results are illustrated by the specific example, providing the valuable insight in the proposed method.

Overall, the manuscript is well written and sound, therefore the suggestion is to be accepted in its present form. 

Author Response

We thank the reviewer for the evaluation of our work.

Since no changes have been requested, we only respond to the comments by Reviewer #2.

Reviewer 2 Report

1. Of great importance is study  on stability of a linearized system.
It is wrong to suppose that infinitesimal perturbations only are taken into account because in fact perturbations depend on values of characteristic parameters.
Evidently, the paper lacks study of Lyapunov instability for linearized systems in which special solutions growing in time are constructed.
This analysis could  help to precise very rough estimations of $Re$ and Wi$ obtained in the paper. 
2. Some notations look inappropriate and make reading of this sufficiently interesting paper difficult.
3. Two sets of numbers $Re$ and $Wi$ are possible for the Taylor-Couette flow in section 6. It is necessary to analyze values of these parameters in the case of Lyapunov stability with  finite perturbations .

Author Response

We thank the reviewer for the evaluation of our work.

Comment #1:

Of great importance is study  on stability of a linearized system. It is wrong to suppose that infinitesimal perturbations only are taken into account because in fact perturbations depend on values of characteristic parameters. Evidently, the paper lacks study of Lyapunov instability for linearized systems in which special solutions growing in time are constructed. This analysis could  help to precise very rough estimations of $Re$ and Wi$ obtained in the paper. 

Response #1:

We agree that the study of the governing equations linearized in the vicinity of the steady state is an important problem, and we have already emphasized this fact in the introduction. However, the manuscript is focused on non-linear (finite amplitude) perturbations in general geometry. (Taylor-Couette flow serves only as an example of the application of the general theory!) As such the manuscript is, in our opinion, rich in content and also long enough. A thorough discussion of the relation between the linearized stability, special (problem specific) growing-in-time solutions, and the general theory for non-linear (finite amplitude) perturbations is definitely needed, but it is beyond the scope of the current manuscript. (And, naturally, the subject of future work.)

Changes in manuscript #1: We have added more references to the papers focused on linearised stability and on special solutions that has been found numerically. See line 24-27 and line 34 in the revised manuscript

Comment #2:

Some notations look inappropriate and make reading of this sufficiently interesting paper difficult.

Response #2:

This comment is difficult to answer, since it is not very specific. What notation exactly seems to be inappropriate? The notation used in the description of the Giesekus model is the same as in the paper Rajagopal, K. R., & Srinivasa, A. R. (2000). A thermodynamic frame work for rate type fluid models. Journal of Non-Newtonian Fluid Mechanics, 88(3), 207-227. This paper seems to be a classical one in the field. (The paper is included in 40th Anniversary Article Collection of the Journal of Non-Newtonian Fluid Mechanics, see https://www.journals.elsevier.com/journal-of-non-newtonian-fluid-mechanics/highlighted-articles/40th-anniversary-article-collection.) The notation for function spaces and norm/metric is in our opinion also standard in the theory of partial differential equations. Furthermore, Reviewer #1 have not commented on the notation at all, hence he/she probably finds it acceptable.

Changes in manuscript #2: We have taken no action regarding the notation.

Comment #3:

Two sets of numbers $Re$ and $Wi$ are possible for the Taylor-Couette flow in section 6. It is necessary to analyze values of these parameters in the case of Lyapunov stability with  finite perturbations.

Response #3: We do not understand the comment. Everything that is done in Section 6 applies to finite perturbations. (In particular, Figure 3 is obtained for finite perturbations.) We are working with the nonlinear governing equations without any simplification.